# Mass concentration estimates of long-range-transported Canadian biomass burning aerosols from a multi-wavelength Raman polarization lidar and a ceilometer in Finland

Xiaoxia Shang[1], Tero Mielonen[1], Antti Lipponen[1], Elina Giannakaki[1,2], Ari Leskinen[1,3], Virginie Buchard[4,5], Anton S. Darmenov[4], Antti Kukkurainen[1,3], Antti Arola[1], Ewan O'Connor[6], Anne Hirsikko[6], Mika Komppula[1]

[1]Finnish Meteorological Institute, Kuopio, Finland
[2]Department of Environmental Physics and Meteorology, University of Athens, Athens, Greece
[3]Department of Applied Physics, University of Eastern Finland, Kuopio, Finland
[4]NASA/Goddard Space Flight Center, Greenbelt, MD, USA
[5]GESTAR/Universities Space Research Association, Columbia, MD, USA
[6]Finnish Meteorological Institute, Helsinki, Finland

*Correspondence to*: Xiaoxia Shang (xiaoxia.shang@fmi.fi)

**Abstract.** A quantitative comparison study for Raman lidar and ceilometer observations, and for model simulations of mass concentration estimates of smoke particles is presented. Layers of biomass burning aerosol particles were observed in the lower troposphere, at 2 to 5 km height on 4 to 6 June 2019, over Kuopio, Finland. These long-range-transported smoke particles originated from a Canadian wildfire event. The most pronounced smoke plume detected on 5 June was intensively investigated. Optical properties were retrieved from the multi-wavelength Raman polarization lidar Polly[XT]. Particle linear depolarization ratios (PDR) of this plume were measured to be $0.08 \pm 0.02$ at 355 nm and $0.05 \pm 0.01$ at 532 nm, suggesting the presence of partly coated soot particles or particles that have mixed with a small amount of dust or other non-spherical aerosol type. The layer-mean PDR at 355 nm (532 nm) decreased during the day, from ~ 0.11 (0.06) in the morning to ~ 0.05 (0.04) in the evening; this decrease with time could be linked to the particle aging and related changes in the smoke particle shape properties. Lidar ratios were derived as $47 \pm 5$ sr at 355 nm and $71 \pm 5$ sr at 532 nm. A complete ceilometer data processing for a Vaisala CL51 is presented from sensor provided attenuated backscatter coefficient to particle mass concentration (including the water vapor correction for high latitude for the first time). Aerosol backscatter coefficients (BSCs) were measured at four wavelengths (355, 532, 1064 nm from Polly[XT], and 910 nm from CL51). Two methods, based on a combined lidar and sun-photometer approach, are applied for mass concentration estimations from both Polly[XT] and the ceilometer CL51 observations. In the first method #1 we used converted BSCs at 532 nm (from measured BSCs) by corresponding measured backscatter-related Ångström exponent, whereas in the second method #2 we used measured BSCs at each wavelength independently. A difference of ~ 12 % or ~ 36 % was found between Polly[XT] and CL51 estimated mass concentrations using method #1 or #2, showing the potential of mass concentration estimates from ceilometer. Ceilometer estimations have uncertainty of ~ 50 % in the mass retrieval, but the potential of the data lays in the great spatial coverage of these instruments. The mass retrievals were compared with the Modern-Era Retrospective analysis for Research and

Applications, version 2 (MERRA-2) meteorological and aerosol reanalysis. The inclusion of dust (as indicated by MERRA-2 data) in the retrieved mass concentration is negligible considering the uncertainties, which also shows that ceilometer observations for mass retrievals can be used even without exact knowledge on the composition of the smoke dominant aerosol plume in the troposphere.

## 1 Introduction

Wildfires release large amounts of aerosols into the atmosphere, contributing significantly to direct radiative forcing (IPCC 2013, https://www.ipcc.ch/site/assets/uploads/2018/02/WG1AR5_Chapter08_FINAL.pdf, last access: 31 March 2021) and affecting cloud optical properties by acting as cloud condensation nuclei (Yu, 2000) or ice nuclei (Prenni et al., 2012). Biomass burning is the dominant global source for carbonaceous aerosols, including organic and black carbon (Andreae, 2019), which can be transported over thousands of kilometres in the atmosphere (Andreae, 1991; Fromm and Servranckx, 2003; Mielonen et al., 2012; Portin et al., 2012). These smoke plumes can mix with other aerosols (such as dust) originating from regional and local sources (Osborne et al., 2019; Tesche et al., 2009). Long-range transport of particles originating from biomass burning has been recognized as a significant source of tropospheric aerosols at northern latitudes (Generoso et al., 2003), with the most common being smoke from Russia or North America (Müller et al., 2005; Wotawa et al., 2001). The optical properties of smoke particles vary depending on the vegetation fuel types in the source regions and the combustion phase; they also change frequently when dispersing in the air (Reid et al., 2005a, 2005b).

Lidars provide quantitative range-resolved information of atmospheric aerosols. Multi-wavelength Raman lidar, together with its depolarization capability, provides comprehensive information on aerosol optical and microphysical properties (Müller et al., 1999, 2005), and allow the identification of the aerosol type using the intensive optical parameters (Groß et al., 2013; Illingworth et al., 2015). Ground-based lidar networks, such as EARLINET (European Aerosol Research Lidar Network, https://www.earlinet.org, last access: 3 May 2021, Pappalardo et al., 2014), PollyNET (Raman and polarization lidar network, http://picasso.tropos.de, last access: 3 May 2021, Baars et al., 2016), and MPLNET (Micropulse Lidar Network, https://mplnet.gsfc.nasa.gov, last access: 20 July 2021, Welton et al., 2001) have continued to provide observations of clouds and aerosols over large spatial scales. Adam et al. (2020) present a methodology for analysing the biomass burning events recorded in the EARLINET database, and provide a literature review of lidar-derived intensive parameters of biomass burning aerosols (46 reference values from 39 cited papers), including fresh and aged ones. Lidar observations showed that biomass burning aerosols are medium- to high-absorbing particles with an almost spherical shape and small particle size, producing medium to high lidar ratios, low depolarization ratios and high Ångström exponents (Alados-Arboledas et al., 2011; Amiridis et al., 2009; Baars et al., 2012; Müller et al., 2007; Murayama et al., 2004; Nepomuceno Pereira et al., 2014). Spaceborne lidars such as CALIOP (Cloud-Aerosol Lidar with Orthogonal Polarization) onboard the CALIPSO (Cloud-Aerosol Lidar and Infrared Pathfinder Satellite Observations) (Winker et al., 2009), and the ADM-Aeolus lidar of European Space Agency (ESA) (Stoffelen et al., 2005) are complementary to these network observations by providing 3-D aerosol

distributions around the globe, which also contribute significantly to the monitoring and documentation of the transport of the smoke (Baars et al., 2021; Kim et al., 2009; Ohneiser et al., 2020).

Numerous studies have investigated the properties of smoke plumes transported from Canadian wildfires to Europe (Ansmann et al., 2018; Fiebig et al., 2003; Hu et al., 2019; Müller et al., 2005). For example, in 2017 a record-breaking Canadian wildfire smoke event was observed over European lidar stations. The arrival of biomass burning smoke layers from this event in August 2017 was first reported by Khaykin et al. (2018). Haarig et al. (2018) present night-time lidar observations of wildfire smoke aerosols during the event in both tropospheric and stratospheric layers over Leipzig, with lidar ratios of 40–45 sr (355 nm), 65–80 sr (532 nm), 80–95 sr (1064 nm), low depolarization ratio (<0.03 at 355, 532, 1064 nm) for plumes in the troposphere and higher depolarization ratio (0.22 at 355nm, 0.18 at 532 nm, 0.04 at 1064 nm) for plumes in the stratosphere. Later on, Baars et al. (2019) reported six months observations (from August 2017 to January 2018) of such wildfire smoke aerosols during the episode with a network of 28 EARLINET ground-based lidars in Europe, showing the aerosol properties and the evolution of the smoke layer during the long-range transport. Recently, wildfire smoke layers were measured over the North Pole with a lidar aboard the icebreaker Polarstern during the MOSAiC (Multidisciplinary drifting Observatory for the Study of Arctic Climate) expedition (Engelmann et al., 2020; Ohneiser et al., 2021). However, the spatial resolution remains sparse, as advanced lidars are expensive. Similar observational records over Northern Europe are more scarce in the literature.

Several national weather services have built up ceilometer networks for cloud monitoring (e.g., http://ceilometer.fmi.fi, Hirsikko et al., 2014, E-Profile: https://e-profile.eu, last access: 21 April 2021) with unattended operation on a 24/7 basis. Information from the large number of ceilometers in these networks can fill the gaps between advanced lidar stations. Ceilometers are single-wavelength, eye-safe backscatter lidars, originally designed to determine cloud base heights. Studies (e.g., Wiegner and Geiß, 2012) show that ceilometers can also be used to retrieve the aerosol backscatter coefficient with high accuracy. However, the accuracy of the aerosol extinction coefficient retrieval is sensitive to the estimate of the unknown lidar ratio (LR). Ceilometers typically operate in the near-infrared (1064 nm or 910 nm) but the lidar ratios for different aerosol types have usually been observed and reported only at 532 and 355 nm. Only recently have lidar ratios at 1064 nm been measured by Raman lidar (Haarig et al., 2016).

Ceilometer measurements have been used in several aerosol studies even though the instruments were originally designed to measure cloud heights. From an Arctic station, Mielonen et al. (2013) reported ceilometer observations of biomass burning plume heights from the 2010 Russian wildfires in northern Finland. Tsaknakis et al. (2011) present an inter-comparison of lidar and ceilometer measurements under different atmospheric conditions (urban air pollution, biomass burning and Saharan dust event), showing good agreements in determining the mixing layer height and the attenuated backscatter coefficient. Cazorla et al. (2017) present the implementation of procedures to manage the Iberian Ceilometer Network (ICENET) for monitoring aerosol characterization for near real time, which has been tested during a dust outbreak. Ceilometer measurements of the German Weather Service (DWD) network (http://www.dwd.de/ceilomap, last access: 20 July 2021) were employed to follow the progression of the volcanic ash layer (Emeis et al., 2011), and to visualise the dispersion and

temporal development of the North American smoke plumes (Trickl et al., 2015). Vaughan et al. (2018) showed how a dense network of lidars and ceilometers in UK tracked the evolution of Canadian forest fire smoke. Adam et al. (2016) demonstrated that the operational ceilometer network of the Met Office can also provide valuable information for monitoring pollution events. Huff et al. (2021) demonstrated that ceilometers in the Unified Ceilometer Network (UCN, https://alg.umbc.edu/ucn/, last access: 20 July 2021) can verify and track smoke plume transport from a prescribed fire, in Maryland. Calibrated ceilometer profiles were also used as a tool to evaluate the aerosol forecasts by the European Centre for Medium-Range Weather Forecasts (ECMWF) Integrated Forecasting System aerosol module (IFS-AER) (Flentje et al., 2021). Dionisi et al. (2018) proposed a model-assisted methodology to retrieve key aerosol properties (such as extinction coefficient, surface area, and volume) from ceilometer measurements, under continental conditions; the good performances of that approach suggest that ceilometers can provide quantitative information for operational air quality and meteorological monitoring. In order to analyse to what extent the existing ceilometer infrastructure could do in case of smoke monitoring, we performed a comparison study using an advanced Raman lidar, a ceilometer, and model data.

On 4–6 June 2019, biomass burning aerosol layers were observed in the lower troposphere over Kuopio, Finland. These smoke particles originated from a Canadian wildfire event. In this study, we present observations of the smoke plume from a multi-wavelength Raman polarization lidar Polly$^{XT}$ and a Vaisala CL51 ceilometer. A combined lidar–photometer approach is presented for estimating mass concentration as a good knowledge of the aerosol mass concentration is required from the aviation safety point of view (Schumann et al., 2011). Based on this approach, we applied two methods in this study: method #1, measured backscatter coefficients were converted to backscatter coefficients at 532 nm by corresponding measured backscatter-related Ångström exponent, and then be applied to estimate the mass concentrations; method #2, mass concentrations were estimated from measured backscatter coefficients at each wavelength (355, 532, 1064 nm from Polly$^{XT}$, and 910 nm from CL51) independently. This study reports, for the first time, a quantitative comparison study of mass concentration estimates of smoke particles, for Raman lidar and ceilometer observations. Moreover, we demonstrate the usefulness of a Vaisala ceilometer to monitor smoke (in terms of quantitative information on the aerosol load) in the troposphere; the potential for mass concentration retrieval from ceilometer observations is also discussed. In addition, the mass retrievals were compared with the Modern-Era Retrospective analysis for Research and Applications, version 2 (MERRA-2) meteorological and aerosol reanalysis.

This paper is organized as follows: in Sect. 2, the measurement site, instrumentation, and data analysis are described. In Sect. 3, the mass estimation methods and results are presented and discussed. The conclusions are given in Sect. 4.

## 2    Measurement site, instrumentation and data analysis

The Vehmasmäki station in Kuopio (62°44'N, 27°33'E; 190 m above sea level), belonging to the European Aerosol Research Lidar Network (EARLINET, https://www.earlinet.org, last access: 3 May 2021) and PollyNET (http://picasso.tropos.de, last access: 3 May 2021), is a rural site, located ~18 km from the city centre of Kuopio, in Eastern Finland. The station has been

an operational profiling site since Autumn 2012 and is operated by the Finnish Meteorological Institute (Hirsikko et al., 2014). It is equipped with a ground-based multi-wavelength Raman polarization lidar Polly[XT] (Baars et al., 2016; Engelmann et al., 2016), Vaisala ceilometers CL31 and CL51, a Doppler lidar, and in situ instruments, next to a 318 m tall mast (for the meteorological observations) during the period. Vehmasmäki is located far from major aerosol sources such as dust or anthropogenic aerosol and the atmosphere is relatively clean.

## 2.1 Ancillary data

The closest AErosol RObotic NETwork (AERONET, http://aeronet.gsfc.nasa.gov, last access: 3 May 2021) station to the Vehmasmäki site is the Kuopio station (62°53'N, 27°38'E; 105 m above sea level), ~18 km from Vehmasmäki. The AERONET sun-photometers measure the aerosol optical depth (AOD) from 340 to 1640 nm (eight channels) for the total atmospheric column, with an uncertainty from 0.01 to 0.02 (Eck et al., 1999). The AERONET (version 3.0) level 2.0 direct sun products (O'Neill, 2003) and inversion products (Dubovik and King, 2000) were used in this study. These products include spectral AODs, fine-mode and coarse-mode AODs (at 440, 675, 870, 1020 nm of inversion products; or at 500 nm of direct sun products), fine-mode and coarse-mode related volume concentrations in the entire vertical atmospheric column. The volume particle size distribution was retrieved in the range of radius of 0.05–15 µm; the minimum within the size interval from 0.439–0.992 µm was used as a separation point between fine and coarse mode particles. A detailed uncertainty analysis was performed by Dubovik et al. (2000), showing low errors in AOD, and about 10 %–20 % error in volume concentration.

Temperature and pressure profiles from the GDAS (Global Data Assimilation System, https://www.ready.noaa.gov/gdas1.php, last access: 19 March 2021) database were used for the correction of Rayleigh extinction and backscattering effects for lidar data analysis. Aiming at the observations of water vapor profiles, the most used and well-established measurement method is radiosonde sounding (Wiegner et al., 2019). However, the closest available radiosonde data are from Jokioinen (Finland), located ~ 300 km away from the measurement site. Filioglou et al. (2017) reported the inadequate vertical representation of water vapor due to the non-stable atmospheric conditions between two sites, when using a radiosonde 100 km away. Thus, the relative humidity profiles from GDAS data were used for the water vapor number densities estimations. HYSPLIT (Hybrid Single Particle Lagrangian Integrated Trajectory, https://ready.arl.noaa.gov/HYSPLIT.php, last access: 19 March 2021) backward trajectories were also analysed to study the air mass origins. Additionally, the thermal anomalies (fire sources) from Terra and Aqua MODIS data (MODIS, 2019) were synergistically used to locate where the forest fires were occurring. The "Dust score" data provided by AIRS (Atmospheric InfraRed Sounder) were used to determine the occurrences of dust events (https://airs.jpl.nasa.gov, last access: 1 July 2021). MERRA-2 is a global reanalysis produced with the NASA global Earth System model, GEOS (Goddard Earth Observing System) coupled with the aerosol module GOCART (Goddard Chemistry, Aerosol, Radiation and Transport) (Gelaro et al., 2017) and includes the assimilation of aerosol observations (AOD) from various spaceborne instruments such as MODIS, AVHRR, MISR (Buchard et al., 2017; Randles et al., 2017). It has an approximate horizontal resolution of 0.5° × 0.625° and

72 hybrid-eta level levels from the surface to 0.01 hPa. Every 3 hours, MERRA-2 produces vertical profiles of aerosol mass mixing ratio for five aerosol species (dust, sea salt (SS), black and organic carbon (BC and OC), and sulfate (SU)) from which lidar optical parameters, such as aerosol extinction and backscattering coefficients can be calculated.

## 2.2 Polly[XT] lidar

The multi-wavelength Raman polarization lidar Polly[XT] has three emission wavelengths (355, 532, 1064 nm) and 12
detection channels, including a far-range receiver unit with eight channels (355, 387, 407, 532, 607, 1064 nm, and two depolarization channels at 355 and 532 nm), a near-range receiver unit with four channels (355, 387, 532, and 607 nm). The initial spatial and time resolution is 7.5 m and 30 s, respectively. The laser beams are tilted to an off-zenith angle of 5° to avoid specular reflections from horizontally aligned ice crystals. For the calculation of optical properties in this study, the profiles were temporally averaged in 2 h intervals, and smoothed with a vertical gliding averaging window length of 11 bins
(a vertical range of ~ 82 m).

Data processing methods of Raman lidars are well established. When the signal-to-noise ratio is high enough for the received signals at inelastic Raman-shifted wavelengths (387 and 607 nm), which is mainly during night-time, profiles of extinction and backscatter coefficients at 355 and 532 nm can be derived independently using the Raman inversion technique (Ansmann et al., 1992a). Otherwise, the Klett–Fernald method (Fernald, 1984; Klett, 1981) is applied using the elastic
signals to retrieve the backscatter coefficients. The relative uncertainties are in the range of 5–10 % for backscatter coefficients and depolarization ratios at 355 and 532 nm (Ansmann et al., 1992b; Baars et al., 2012). The backscatter coefficients retrieval at 1064 nm may be possible with a relative uncertainty of 15 % using only elastic signal by assuming a proper lidar ratio. The lidar ratios at 355 and 532 nm are measured with a typical relative uncertainty of ~ 20 % when the inelastic measurements are good enough. Higher uncertainties in lidar ratio at 1064 nm (~ 30 %) should be considered
(Haarig et al., 2018). Further details on the instrument setup, principle and error propagation can be found in Engelmann et al. (2016).

## 2.3 Ceilometer and data processing

The Vaisala CL51 ceilometer used in this study is a commercial elastic backscatter lidar originally intended for measuring cloud base heights. It operates at 910 nm with an initial temporal resolution of 10 s and range resolution of 10 m, and was
tilted to an angle of 12°–13° from vertical. The horizontal distances between the laser beams of CL51 and Polly[XT] are of the order of ~ 700 m at 5 km, which is considered negligible in this study. The signal-to-noise ratio (SNR) for raw CL51 backscatter signals above the boundary layers is weak, hence some temporal averaging and vertical smoothing were required when performing further analysis. In this study, CL51 signals were smoothed with a vertical gliding averaging window length of 7 bins (a vertical range of ~ 70 m). The profiles were temporally averaged in 10 min intervals for the time–height
cross section quick look, and in 2 h intervals to calculate the optical properties.

Kotthaus et al. (2016) states that background corrections are needed for some ceilometer firmware versions. Following the method proposed in Kotthaus et al. (2016), range histograms of observations from CL51 on clear-sky days were analysed. The results show that the background corrections are not needed for the CL51 data from Kuopio station, because the CL51 is operating with a specific firmware version which is recommended by E-Profile and ACTRIS (the Aerosol, Clouds and Trace
Gases Research Infrastructure, https://www.actris.eu, last access: 3 May 2021).

The instrument constant is not accurately calibrated in absolute terms for many of the ceilometers in the network (http://ceilometer.fmi.fi, last access: 3 May 2021), because the main application is the cloud-base height detection in which a correct instrument constant is not required. Different calibration procedures (e.g., relative/absolute calibration) have been proposed and applied in literature. The stratocumulus cloud technique (O'Connor et al., 2004) is the most appropriate for the
CL51 sensor which was used in this study and recommended by E-Profile for this sensor type. Stratocumulus cloud cases in 2019 were analysed, and five good cases from April to September 2019 were found. A calibration factor of $1.75 \pm 0.03$ was derived with small standard deviation during the 6 months, showing that the CL51 system is quite stable. A relative uncertainty on the instrument constant within 10 % should be considered, as the calibration approach contains a range of possible multiple-scattering factors.

Wiegner and Gasteiger (2015) report that the ceilometer signal must be corrected for water vapor if emitting wavelengths around 910 nm are used. They show that the error in the backscatter coefficient retrieval can be in the order of 20 % for mid-latitudes, and more than 50 % for the tropics, if water vapor absorption is ignored. We performed similar simulations in Kuopio station, following the method given in Wiegner and Gasteiger (2015). The water vapor number densities were calculated from the relative humidity and the temperature profiles from GDAS data. The water vapor absorption cross-
sections in the spectral range between 900 and 920 nm were simulated based on HITRAN (Rothman et al., 2005) data base, which covers a wide range between $10^{-28}$ cm$^2$ and $10^{-21}$ cm$^2$. Wiegner and Gasteiger (2015) state that the annual variability of pressure and temperature has no significant influence on the water vapor absorption cross-sections. It is possible to use the tabulated mean absorption cross-section to calculate an approximative water vapor transmission with a high accuracy (the inherent error of the squared water vapor transmissions is <0.3 %; more details are given in section 4 in Wiegner and
Gasteiger, 2015). For the CL51, we assumed a Gaussian shape of the spectrum, with the central wavelength ($\lambda_0$) of 910 nm and a full width at half maximum of 3.4 nm, as specified by Vaisala. Thus, the water vapor transmission can be estimated, and the effective water vapor transmission was applied to the ceilometer signals for the water vapor corrections (equations and more details can be found in Wiegner and Gasteiger, 2015). One example is given in Fig. 1:

(I) Using the forward integration method, the retrieved particle backscatter coefficients without water vapor
correction ($\beta^*$) were much lower than the ones with water vapor correction ($\beta$) (Fig. 1c). This underestimate increased with height in the boundary layer, and with ~ 40 % for the elevated layer.

(II) Using the backward integration method, neglecting the water vapor led to an overestimate, which increased with the distance from the chosen reference height. An overestimate of ~ 50 % can be found at near ground for the

given example. Nevertheless, a much smaller error (~ 6 %) was found for the elevated layer using this method
compared to the forward integration method.

The uncertainties range due to wrong assumptions of $\lambda_0 \pm 2$ nm is given by the horizontal lines in Fig. 1. The uncertainties in backscatter coefficients of the analytical solution were also shown by dashed lines. Bedoya-Velásquez et al. (2021) applied a water vapor correction method on the CL51 ceilometer measurements, based on the one proposed by Wiegner and Gasteiger (2015). They have also studied the sensitivity of the aerosol retrievals to the use of modelled temperature and absolute
humidity from HYSPLIT to correct water vapor absorption, instead of the co-located Microwave radiometer measurements: it leads to errors in the pre-processed range-corrected signals up to 9 %, and in particle backscatter coefficients up to 2.2 %. Thus, an extra uncertainty should be considered as GDAS data were used for the water vapor correction. We cannot quantify the error in GDAS temperature. Nevertheless, Polly$^{XT}$ measured relative humidity (RH) were applied for the comparison, and good agreements were found. The relative difference on the squared effective water vapor transmissions using RH
profiles of GDAS or Polly$^{XT}$ is less than 2 % for the case in Fig. 1. The input uncertainty in water vapor transmissions due to the use of modelled input was not taken into account in this study, as there was no means to quantify the value. As the water vapor contribution cannot be neglected at Kuopio during summer, the water vapor corrections have been applied to CL51 data in this study.

The retrieval methods for deriving the backscatter coefficient from ceilometers are quite mature (Wiegner et al., 2014;
Wiegner and Geiß, 2012). Under favourable conditions, a relative error of the backscatter coefficient on the order of 10 % seems feasible with a careful calibration by applying the forward integration. On the contrary, significant temporal averaging of ceilometer data is required for performing a Rayleigh calibration, as the detection of molecular signals is intrinsically very difficult. Binietoglou et al. (2011) propose a two-step approach, resulting promising agreement comparing to their lidar PEARL (Potenza EARLINET Raman lidar). The uncertainty of the backscatter coefficient could be in the range of 20–30 %
using the backward integration. The advantage of the forward algorithm is that calibration is required only occasionally, and it is not affected by the low SNR in the upper troposphere. However, the accuracy in deriving extinction coefficients is limited due to the unknown LR at 910 or 1064 nm and its uncertainties. In particular the presence of multi-layered aerosol distributions (with different aerosol types) may introduce more uncertainties. In addition, the uncertainty due to the neglecting the water vapor increased with the distance from the chosen reference height. In this study, we applied the Klett
method (Wiegner et al., 2014) by defining the reference height as close as to the layer of interest, so that the error propagation (due to uncertainties of LR and water vapor transmission) would be minimized for that layer. Characteristic LR values for aerosol types are often measured at 355 or 532 nm; it is only recently that Haarig et al. (2016) measured the LR at 1064 nm, and reported values of 80–95 sr for aged biomass burning smoke (Haarig et al., 2018). A value of 82 sr for LR, as measured at 1064 nm ($82 \pm 27$ sr in Haarig et al., 2018), was assumed as being appropriate for use at 910 nm in this study.

## 3    Results and discussion

From 4 to 6 June 2019, several lofted aerosol particle layers were detected with Polly[XT] (near-real-time quick-looks are publicly accessible at the PollyNET website: http://picasso.tropos.de, last access: 3 May 2021). The AERONET sun-photometer observed enhanced AOD values for these days: the total AOD at 500 nm ranged from 0.24 to 0.42, whereas the mean values for the previous week and the following week were both below 0.08.

In this study, we focus on the observations on 5 June, when the most pronounced aerosol layers were detected. Range-corrected signal (RCS) at 1064 nm from Polly[XT] and at 910 nm from CL51 on 5–6 June are presented in Fig. 2. A dense lofted aerosol layer was highlighted by the enhanced lidar signals, located at ~ 5 km in the morning and which descended to ~ 2 km in the evening; this layer is defined as SPoI (Smoke Plume of Interest). Two faint lofted thin layers were also detected below this layer in the morning. Our in situ pollen measurements (more information about pollen instruments can be found in Bohlmann et al., 2021) shows high pine pollen loading at the ground (highest 2 h pollen concentrations were ~ 3000 m$^{-3}$ on 5 June and ~ 7000 m$^{-3}$ on 6 June). The pollen particles were well mixed in the boundary layer (below 2 km), causing strong backscattering together with high depolarization ratio at 532 nm with a clear diurnal cycle. Although also of interest, the analysis of the pollen layer is out of the scope of this paper.

As shown in Fig. 3, the AERONET (level 2.0 aerosol spectral deconvolution algorithm (SDA) products) fine-mode AOD fraction on 5 June was higher than 93 %, and the Ångström exponent at 500–1020 nm (380–500 nm) varied between 1.4 and 1.8 (0.8 and 1.1), indicating the presence of fine particles in the atmospheric column. The coarse-mode AOD slightly increased during the daytime (always below 0.02) on 5 June, which can be interpreted as the pollen-related contribution to optical depth in the boundary layer. The higher coarse-mode AODs on 6 June could also be a consequence of higher pollen contributions. As a consequence, ceilometer signals were almost totally attenuated above 3 km on 6 June (Fig. 2b).

The backward trajectory analysis was performed using the HYSPLIT model. The analysis shows that particles in the SPoI had travelled about seven days from the forest fire sources (MODIS, 2019) in western Canada to North Europe (Fig. 4). The AIRS dust score map (https://airs.jpl.nasa.gov/map/, last access: 1 July 2021) also showed some dust presence in North America on 30 May.

### 3.1    Optical properties

The pronounced smoke layer, i.e. SPoI shown in black box in Fig. 2, had a layer depth of ~ 1.8 km in the early morning, and became much thinner when descending at night. The Polly[XT]-derived AOD at 355 nm (532 nm) of this layer decreased from 0.21 (0.13) in the morning to 0.04 (0.02) in the night.

Layer-mean values of optical properties of the SPoI were derived and are given in Table 1. Two-hour time-averaged, and vertical smoothed (with a smoothing window of ~ 82 m) lidar profiles were used in order to increase SNR. LR retrievals at 355 nm were available for the whole day, providing layer-mean values of 47 ± 5 sr which remained quite constant during the period. LR retrievals at 532 nm were only possible after sunset (after 18h UTC), resulting in layer-mean values of 71 ± 5 sr.

The 532 nm lidar ratio for aged smoke is larger than the 355 nm lidar ratio, in contrast to urban haze and fresh smoke, where the 355 nm lidar ratio is typically larger than the 532 nm lidar ratio (e.g., Nicolae et al., 2013; Pereira et al., 2014). Actually, as stated by Ansmann et al. (2021) and references therein, this characteristic ratio in LR, i.e. LR(355 nm) / LR(532 nm) <1, is not produced by any other aerosol type and allows a clear identification of aged smoke.

The backscatter coefficients of aged wildfire smoke show a clear and strong wavelength dependence for both 355–532 nm and 532–1064 nm wavelength ranges; the backscatter-related Ångström exponent (BAE) between 355 and 532 nm (between 532 and 1064 nm) shows high values of ~ 2.5 (~ 2.2). Nonetheless, the wavelength dependence of the extinction coefficient for the 355–532 nm spectral range is much weaker, with an extinction-related Ångström exponent (EAE) of ~ 1.4. Nicolae et al. (2013) state that the EAE can be used for identifying the evolution of ageing processes of biomass burning aerosol, as it decreased from 2 for fresh to ~ 1.4–0.5 for ages biomass burning aerosols. The microphysical analysis was not performed in this study; yet the measured EAE would be related to the effective radius of ~ 0.23 µm, when considering the relationship between EAE and effective radius of forest fires smoke reported by Müller et al., (2005) (c.f., Fig. 6 in that paper). This estimated effective radius value is consistent with those of aged smoke aerosols reported in literature (Table 1). The AERONET sun-photometer Ångström exponent at 380–500 nm on 5 June showed lower values than lidar EAE at 355–532 nm; possible cirrus contamination could partly explain as sun-photometer data are for the total atmospheric column. Note that lidar's EAE was only available for night-time (between 5th and 6th).

The smoke particles caused slightly enhanced particle linear depolarization ratios (PDR) at 355 nm (532 nm) with a mean value of 0.08 ± 0.02 (0.05 ± 0.01) in the smoke layer, suggesting the presence of partly coated soot particles or particles that have mixed with a small amount of dust or other non-spherical aerosol type. The layer-mean PDR at 355 nm (532 nm) decreased during the day, from ~ 0.11 (0.06) in the morning to ~ 0.05 (0.04) in the evening. The decrease of the PDR with time could be linked to the particle aging and related changes in the smoke particle shape properties, as stated by Baars et al. (2019). The relative humidity (RH) profiles from GDAS data showed low values in the lower atmosphere (<60 % below 6 km) before 15h UTC, and even lower RH (<40 %) at the SPoI altitude. RH slightly increased in the evening. The signal in the 407 nm Raman-shifted channel was used to determine the water vapor mixing ratio profile during night-time, showing that the layer-mean RH changed from ~ 27 % at 19 h to ~ 38 % at 23 h, which was associated with the advection of a moister air mass with a water vapor mixing ratio close to 1–3 g kg$^{-1}$. The smoke particles were dry, and then captured water vapor in the atmosphere during the evening. The decreasing temperature and increasing RH also increase the probability that smoke particles become glassy. The depolarization ratios of aged biomass burning aerosols (originating from Canada and/or North America) reported in the literature (Table 1) range from 0.01 to 0.11 (0.01 to 0.08) at 532 nm (355 nm). More information of the aged smoke from other regions can be found in the literature review by Adam et al. (2020, see the Supplement).

## 3.2 Mass concentration estimation

Ansmann et al., (2011) present a combined lidar-photometer method that enables the retrieval of the vertical profiles of ash and non-ash (fine-mode) particle mass concentration ($m$). It is based on the mass density ($\rho$) of considered particles (mainly

from literature), the volume-to-extinction conversion factors $c_v$ (from literature or computed from AERONET sun-photometer products), the backscatter coefficient (BSC, $\beta$) (from lidar measurements), and the lidar ratio ($LR$) (from Raman lidar measurements or assumptions depending on aerosol types), following the equation:

$$m_a = \rho_a \cdot c_{v,a}(\lambda) \cdot \beta_a(\lambda) \cdot LR_a(\lambda) \,, \tag{1}$$

where $a$ indicates the aerosol type, and $\lambda$ the wavelength. $c_{v,a}$ uses the temporal mean value within a given time period to convert particle extinction coefficients into particle volume concentrations:

$$c_{v,a}(\lambda) = \overline{\left(\frac{v_a}{\tau_{a(\lambda)}}\right)} \,. \tag{2}$$

The column particle volume concentration $v$ and corresponding optical thickness $\tau$ for aerosol component $a$ are obtained from AERONET sun-photometer products.

This approach was applied to both Polly$^{\text{XT}}$ and CL51 data to estimate the mass concentration profiles for biomass burning aerosols in the SPoI. Adapting from the methods describe by Ansmann et al. (2021), we applied two methods in this study:

Method #1: Mass concentrations were estimated from the measured backscatter coefficients which were converted to 532 nm, using the corresponding measured backscatter-related Ångström exponent. The volume-to-extinction conversion factors at 532 nm from literature was applied (currently the only available wavelength for the smoke factor in the literature).

Method #2: Mass concentrations were estimated from measured backscatter coefficients at each wavelength of 355, 532, 1064 and 910 nm. The volume-to-extinction conversion factors were evaluated at corresponding wavelengths using AERONET data.

In this study, we assume that both methods can be applied appropriately, and the limitations and sources of uncertainties of method #2 will be discussed in Sect. 3.2.2. The recommendation on the chosen method will be discussed later.

We assume that there are only biomass burning aerosols in the SPoI. Values for the smoke particle density vary in the literature (Chen et al., 2017; Li et al., 2016; Reid et al., 2005a), but should be in the range of 1.0–1.3 g cm$^{-3}$ (Ansmann et al., 2021). In this study, a particle density $\rho_s$ of 1.3 g cm$^{-3}$ was used for the biomass burning particles. In the SPoI, the backscatter coefficient of smoke particles is assumed to be equal to the total particle backscatter coefficient ($\beta_s = \beta_{total}$). Figure 5 (a) shows the lidar-derived backscatter coefficients at 355 nm (blue), 532 nm (green), and 1064 nm (red) from Polly$^{\text{XT}}$, and at 910 nm (black) from CL51 for two-hour time-averaged lidar profiles on 5 June 2019. Signals were smoothed with vertical gliding averaging window lengths of 11 bins for Polly$^{\text{XT}}$ and 7 bins for CL51. The peak value of backscatter coefficients in the SPoI reached ~ 5 Mm$^{-1}$ sr$^{-1}$ at 355 nm in the morning (6–8h UTC) and reduced to ~ 3 Mm$^{-1}$ sr$^{-1}$ at 355 nm at night (22–24h UTC).

### 3.2.1   Method #1: based on BAE & the conversion factor from literature

Ansmann et al. (2021) recommended the used of $0.13 \pm 0.01 \times 10^{-6}$ m as the conversion factor at 532 nm for the smoke observations far away from fire regions. In this section, we used this recommended smoke conversion factor of 0.13 (denoted as $c_v^{532}$). The backscatter-related Ångström exponents (BAE) between different wavelength ($\lambda$, i.e., 355, 1064 nm of Polly$^{\text{XT}}$, 910 nm of CL51) and 532 nm (of Polly$^{\text{XT}}$) were derived (Eq. 3) for the SPoI (shown in Table 2) using the measured backscatter coefficients (BSC, $\beta(\lambda)$). These measured backscatter coefficients were converted to the wavelength

of 532 nm, denoted as $\beta_{conv}^{532}$, following the Eq. (4); the profiles of these converted backscatter coefficients are given in Fig. 5 (b). The mass concentration can be thus estimated by Eq. (5), as the product of $\beta_{conv}^{532}$, the smoke lidar ratio at 532 nm ($LR^{532}$), the conversion factor at 532 nm, and the smoke particle densities (Table 2). The mass concentration profiles of the SPoI, retrieved from measured backscatter coefficients at four wavelengths based on method #1, are given in Fig. 5 (c).

$$BAE(\lambda, 532) = -\frac{ln\left(\frac{\beta(\lambda)}{\beta(532)}\right)}{ln\left(\frac{\lambda}{532}\right)} \tag{3}$$

$$\beta_{conv}^{532}(\lambda) = \beta(\lambda) \cdot \left(\frac{532}{\lambda}\right)^{-BAE(\lambda,532)} \tag{4}$$

$$m = \rho \cdot c_v^{532} \cdot LR^{532} \cdot \beta_{conv}^{532}(\lambda) \tag{5}$$

The peak value of the mass concentrations was found at 6–8h UTC, of ~ 23.5 (27.5) μg m$^{-3}$ estimated from the backscatter coefficients at 532 nm (910 nm). If we take the mass concentration estimated from the BSC at 532 nm as the reference, good agreements are found between the mass concentrations estimated from BSCs at different wavelengths (Fig. 5 d). The mean

values of the relative differences were around 8 %, 12 %, and 18 % for the estimations from BSCs at 355, 910 and 1064 nm, respectively. Comparing 532 and 355 nm mass estimates, better agreements were found during daytime (8–20h UTC), with a difference <6 %. Nonetheless, considering 532 and 910 nm estimates, the best agreements were found at 6–8 and 20–24h UTC, with a difference <3 %, whereas the worst agreement of ~ 30 % was found at 14–16h UTC. Larger differences between 910 and 1064 nm estimates were found, with a mean relative difference of ~ 28 %, and a highest value of ~ 64 % at

14–16h UTC.

In Table 3, the uncertainties in the input parameters and the estimated mass concentrations are listed. We assume an uncertainty of 20 % in the smoke mass density (Ansmann et al., 2021). The uncertainties in backscatter coefficients at different wavelengths and lidar ratio at 532 nm follow from the discussions in Sect. 2. The conversion factor and lidar ratio at 532 nm are required as input, with assumed uncertainties of 10 % (given in Ansmann et al., 2021) and 20 % (c.f., Sect.

2.2), respectively. The uncertainties in BAE between different wavelength pairs, and in $\beta_{conv}^{532}$ were obtained by error propagations to Eqs. (3,4). Note that the standard deviations of BAE from our measurements (Table 2) show lower values than their uncertainties. Finally, after applying the law of error propagation to Eq. (5), we expect an overall uncertainty in the

mass concentration estimates of 32–45 %. The highest uncertainty of 45 % was found when using the ceilometer method, mainly due to the higher uncertainty of 20 % in the backscatter coefficient retrieval.

However, the lidar measurements at 532 nm are not always collocated, especially for numerous ceilometer stations. For those cases, the lidar ratio at 532 nm and the BAEs (or colour ratios) should be assumed, thus with higher uncertainties. We can assume uncertainties of 30 % in lidar ratio at 532 nm and 30 % in BAEs for all wavelength pairs, thus, the uncertainty for the estimated mass concentrations will be over 50 % (Table 2). For the smoke particles, extended overviews of observed wavelength dependencies of backscatter coefficients can be found in Burton et al. (2012) and Adam et al. (2020).

### 3.2.2 Method #2: BSC at each wavelength & conversion factors from site

The method #1 is recommended when the measurements at 532 nm are additionally available, or the BAE (or backscatter colour ratio) can be reasonably assumed. Nevertheless, here we suggest a second method, in which mass concentrations were estimated from measured backscatter coefficients at several wavelengths independently, and the measurement at one single wavelength (e.g., for elastic lidars and ceilometers) is required as input for each estimate. This method #2 is recommended in the regions with the pure aerosol type (dust, smoke, marine, etc) condition, where the conversion factor can be evaluated with high accuracy. The mass estimations of the SPoI from measured backscatter coefficients at each wavelength are compared in this section.

Since AERONET inversion products of level 2.0 were not available on 5 June 2019, the AERONET products for the observations on 6 June 2019 (three distributions at 06:33, 07:10, and 14:41) were used to compute the volume-to-extinction conversion factors at different wavelengths for fine-mode particles ($c_v$). The fine-mode fraction on 6 June was a bit lower than on 5 June (Fig. 3), but still higher than 90 % before 10h UTC and around 87 % in the afternoon (11–16h). From the size distribution, the separation points between fine and coarse mode particles were found as ~ 0.576 µm (the size classes 1–10 were considered for fine-mode aerosols). Here, the assumption is made that the photometer-derived fine particles are mainly smoke particles. Both the Dubovik approach (Dubovik et al., 2006; Dubovik and King, 2000) and the O'Neill method (2003) were applied at the wavelength of 532 nm, resulting in similar values for this factor (~ 0.1 % difference). For wavelengths of 355, 910 and 1064 nm, only the Dubovik approach was applied. The mean conversion factors at four wavelengths are given in Table 2 (method #2) together with their standard deviations.

The estimated conversion factor value at 532 nm of 0.211 ± 0.003 ×10$^{-6}$ m is higher than what we used in the previous section, with the difference ($\Delta c_v$=0.08 ×10$^{-6}$ m) larger than the uncertainty. This value is higher than the values for both fresh and aged smoke observations (from 0.13 ± 0.01 to 0.17 ± 0.02 ×10$^{-6}$ m) at several AERONET stations reported in Ansmann et al. (2021). However, Ansmann et al. (2012) also applied a high value of 0.24 ± 0.02 ×10$^{-6}$ m for the mass concentration retrieval of smoke aerosols (fine mode) when studying lofted layers containing desert dust and biomass burning smoke. It is hard to distinguish between smoke and urban haze aerosols, as they are often small (with size up to about 1 µm in radius) and quasi-spherical aerosols. Further, the characteristic conversion factors are in the similar value range. For examples,

Ansmann et al., (2011) reported a conversion factor of $0.18 \pm 0.02 \times 10^{-6}$ m for the central European haze; Mamali et al., (2018) found a factor of $0.14 \pm 0.02 \times 10^{-6}$ m for continental/pollution particles over Cyprus; Mamouri et al., (2017) computed a factor of $0.30 \pm 0.08 \times 10^{-6}$ m for continental aerosol pollution over Germany.

Air mass sources of aerosols on 6 June were investigated by the backward trajectory analysis (HYSPLIT model). It shows that some of the particles were coming from the forest fire in Canada region, while part of them were transported from

Poland where urban haze could have been with smoke aerosols (e.g., Fig. 6). The aerosol subtype products (version 4.20) from CALIPSO when the orbit passing over Poland on 3 June (orbit from UTC 11:44 to 11:58) and 4 June (orbit from UTC 01:18 to 01:31) indicate the presence of polluted continental/smoke and polluted dust.

Consequently, it is possible that European pollution was mixed with Canadian smoke aerosols on 6 June in the fine-mode particles. Hence, the retrieved conversion factors cannot perfectly describe the smoke. However, in this section we still

assume these factors reflect the smoke, so as to do the comparison analysis of estimated mass concentration from Polly$^{XT}$ and CL51.

The mass concentrations were derived by the product of the backscatter coefficients (BSCs, $\beta(\lambda)$) at four wavelengths, their respective smoke lidar ratios, the related conversion factors, and the smoke particle densities (Table 2, Eq. 1). The estimated mass concentration profiles are given in Fig. 7 (a), based on the lidar-derived BSCs at 355, 532, 910 or 1064 nm,

independently.

The peak value of the mass concentrations estimated from the BSCs at 532 nm reached $\sim$ 38 µg m$^{-3}$ at 6–8h UTC, higher than the one estimated from method #1 because of the bigger conversion factor. The relative differences on the mass concentrations estimated from the BSCs at different wavelengths were analysed (Fig. 7 b). Similarly, we take the mass concentration estimated from the BSCs at 532 nm (which is the wavelength most often used in earlier studies) as the

reference, and found an underestimate when using BSCs at 355 nm, with a mean bias of $\sim$ 15 %, and a peak bias of $\sim$ 25 % at 4–6h UTC; the best agreement was found for night-time measurements (20–24 h UTC) with a bias <5 %. Nevertheless, an overestimate was found for the mass concentration estimated from the BSCs at 910 nm, with a mean bias of $\sim$ 36 %, a peak bias of $\sim$ 68 % at 14–16h UTC, and a minimum bias of $\sim$ 14 % at 10–12h UTC. The overestimate for CL51-derived mass concentrations could be due to an overestimate of LR at 910 nm, since we used LR at 1064 nm in the calculations. In

addition, big differences (with a mean value of $\sim$ 42 %) were found between the CL51-derived mass concentrations and the ones estimated from the Polly$^{XT}$-derived BSCs at 1064 nm; highest discrepancy were found of $\sim$ 95 % at 14–16h and $\sim$ 75 % at 16–18h UTC, whereas better agreements were found at 4–6h, 10–12h, and 18–24h, with bias <7 %.

The uncertainties in the input parameters and the estimated mass concentrations of this method #2 are listed in Table 3. The uncertainties in the conversion factors from the standard deviation in Table 2 are very small due to the limited sample

number, thus 0.10 was used as proposed in Ansmann et al. (2021). The uncertainties in backscatter coefficients and lidar ratios at each wavelength follow from the discussion in Sect. 2. Uncertainties in the lidar ratio at ceilometer wavelengths are much larger, particularly, as we applied the lidar ratio value measured at 1064 nm to the ceilometer wavelength of 910 nm.

Thus, we assume an uncertainty of 40 % in the ceilometer lidar ratio. The overall uncertainties in the mass concentration estimates are of about 30–50 %, with the highest uncertainty of 52 % when using ceilometer measurements.

As can be seen in Table 3, when applying the method #2, the uncertainty in mass concentration estimations is slightly lower using measured BSCs at 355 nm, whereas higher uncertainties were found when using measured BSCs at 1064 and 910 nm. The main reason lies in the high uncertainties in lidar ratios at 1064 and 910 nm. Hence, when the lidar ratio can be measured or properly estimated, and the conversion factor can be estimated under the pure aerosol type condition, method #2 is recommended. Otherwise, method #1 can be applied by using properly estimated BAEs or colour ratios.

The good agreement between mass concentrations derived from Polly$^{XT}$ and CL51 measurements in this study show the potential of mass concentration estimates from ceilometer. However, when deriving parameters such as mass concentration from ceilometer, the applied parameters (i.e., $BAE_{532,910}$ value for method #1, or $LR_{910}$ and $c_{v,910}$ values for method #2) and their uncertainties should always be carefully evaluated and provided, as the accuracy of the retrieved mass concentration depends primarily on the accuracy of the parameters that are not derived from the ceilometer observations.

### 3.3 Comparison with MERRA-2 model - wildfire smoke and dust aerosol mixture

The mass concentrations from MERRA-2 model data are used for the comparison with the lidar retrievals. An interesting feature in the MERRA-2 simulation results is the presence of dust in the SPoI. The contribution of dust to the total AOD is very low (much lower than the carbon optical depth), indicating that the dust particles are in the fine mode. However, the dust contribution to the total mass concentration is non-negligible. Low values of lidar-derived depolarization ratio suggest
no significant presence of non-spherical particles, but in principle, a small amount of dust could be mixed with the smoke. It is possible that there are biomass burning aerosols and fine dust aerosols in the SPoI, as only fine dust particles should be able to remain long enough in the atmosphere to be transported from North America to Kuopio. Furthermore, the air masses in SPoI passed by the area in North America where dust was present (shown by the AIRS data).

In this section, the MERRA-2 mass concentrations were compared with the mass concentrations estimated from the Polly$^{XT}$
backscatter coefficients at 532 nm (from method #1-Fig. 5 c, and method #2-Fig. 7 a). Note that the main difference on Polly$^{XT}$-estimated mass concentrations from two methods are due to the different conversion factor values (Table 2), thus the mass concentrations estimated from BSCs at 532 nm using method #1 are ~ 40 % lower than method #2. When the Polly$^{XT}$ estimates from method #1 were used as the reference, good consistencies were found in the morning (at 6h, 9h, and 12h UTC), with overestimations (<30 %) of MERRA-2 mass concentrations; whereas large discrepancies were found in the
afternoon, with high overestimations of ~ 160 % at 15h UTC and ~ 90 % at 18h UTC. If the Polly$^{XT}$ estimates from method #2 were used as the reference, good consistencies were also found in the morning (at 6h, 9h, and 12h UTC), but with underestimations (<30 %); and a large overestimation of ~ 63 % was found at 15h UTC. At 15h UTC, the MERRA-2 simulated dust mass concentration fraction is more than half of the MERRA-2 simulated total mass concentration. It is good to keep in mind that both observations and simulations have significant uncertainties. The presence of cirrus cloud in the

upper atmosphere during the day may also have some impacts on MODIS AOD, which is assimilated by the MERRA-2 model.

In order to check how the inclusion of dust (as indicated by MERRA-2) would affect the mass concentration estimations, we assume that there were wildfire smoke and fine dust aerosol mixture in the SPoI. The POLIPHON (Polarization lidar photometer networking) method (Mamouri and Ansmann, 2014, 2017) was applied to separate fine dust (particles with

radius < 500 nm) and biomass burning aerosols for the SPoI. Here we used the Polly$^{XT}$ retrieved particle backscatter coefficient and particle linear depolarization ratio profiles at 532 nm, because the uncertainty in the 355 nm particle depolarization ratios are much larger (Mamouri and Ansmann, 2017). The depolarization ratios at 532 nm of smoke and fine dust particles were assumed to be 0.03 (Haarig et al., 2018) and 0.16 (Sakai et al., 2010). The fine dust and smoke extinction coefficients were obtained by multiplying the backscatter coefficients with their respective lidar ratios as follows: Ansmann

et al. (2019) report that the typical dust lidar ratio is 40 sr at 532 nm, and lidar ratios for fine- and coarse-mode dust were assumed to be the same. For smoke particles, we took the lidar ratio of 71 sr, which was retrieved during our night-time measurements. For the fine dust, the conversion factor $c_{v,d}$ of 0.22 ×10$^{-6}$ m (Ansmann et al., 2019), and the particle density of 2.6 g cm$^{-3}$ (Ansmann et al., 2012) were used. For smoke particles, 1.3 g cm$^{-3}$ was used as the particle density (Ansmann et al., 2021; Reid et al., 2005a), whereas the conversion factors $c_{v,s}$ of 0.13 ×10$^{-6}$ m (method #1) and 0.21 ×10$^{-6}$ m (method #2)

were both applied. These parameters are reported in Table 4. The fine dust and smoke particle mass concentrations were derived using Eq. (1). For the example given in Fig. 8, the fine dust contributes ~ 13 % to the extinction in the SPoI, whereas its mass concentration contributes ~ 32 % (method #1) or ~ 23 % (method #2) to the total mass concentration. However, the derived total mass concentration considering a fine dust and smoke mixture is only ~ 18 % (method #1) or ~ 4 % (method #2) higher than one assuming smoke particles only. The inclusion of a dust mixture results in slightly higher estimated mass

concentration values, with a difference negligible considering the uncertainties.

We have also performed POLIPHON considering coarse mode dust mixture; higher (~20–30 %) total mass concentrations were retrieved but still within the uncertainty range. The aged smoke aerosols may also introduce enhanced depolarization ratios. If we use a bigger value (e.g., 0.05) instead of 0.03 as the smoke depolarization ratio in POLIPHON, the dust impacts on the mass concentration estimations are even smaller. Hence, the mass estimations of the SPoI considering only smoke are

good enough even if the plume contains small amount of dust.

Similar conclusion can also be applied to ceilometer observations. It is not possible to perform the aerosol separation using ceilometer data alone, as no depolarization information is available at this wavelength. For this instrument, only one aerosol type should always be assumed in the layer of interest, which then imparts an additional bias when estimating the mass concentration. However, we have shown in this section that ceilometer observations for mass retrievals can be used even

without exact knowledge on the composition of the smoke plume in the troposphere.

## 4 Summary and conclusions

On 4–6 June 2019, aerosol layers arising from biomass burning were observed in the lower troposphere between 2–5 km in altitude over Kuopio, Finland. Enhanced backscattered signals were detected by both a multi-wavelength Raman polarization lidar Polly$^{XT}$ and a Vaisala CL51 ceilometer. The HYSPLIT backward trajectories analysis and MODIS fire data suggested that these long-range-transported smoke particles originated from a Canadian wildfire event. An AERONET sun-photometer located in Kuopio observed enhanced AOD values in concert with high Ångström exponents, indicating the presence of fine-mode dominant aerosols in the atmospheric column.

The most pronounced smoke plume, defined as SPoI (Smoke Plume of Interest), detected on 5 June was intensively investigated. Lidar ratios were derived from the Raman lidar, as $47 \pm 5$ sr at 355 nm and $71 \pm 5$ sr at 532 nm, showing that the aerosols of biomass burning origin in the SPoI were medium- to high- absorbing particles. Particle linear depolarization ratios in this layer were measured as $0.08 \pm 0.02$ at 355 nm and $0.05 \pm 0.01$ at 532 nm; which could indicate the presence of irregular-shaped aged smoke particles and/or mixing with a small amount of fine dust particles. Complete processing steps for Vaisala CL51 ceilometer data analysis were firstly reported in this study. The water vapor correction was analysed and applied at a high latitude for the first time, showing that water vapor absorption cannot be neglected for high latitude stations during summer. Two methods, based on a combined lidar and sun-photometer approach (based on AERONET products), were applied to both Polly$^{XT}$ and CL51 data for estimating mass concentrations: method #1, measured backscatter coefficients were converted to backscatter coefficients at 532 nm by corresponding measured backscatter-related Ångström exponent, and then be applied to estimate the mass concentrations; method #2, mass concentrations were estimated from measured backscatter coefficients at each wavelength (355, 532, 1064 nm from Polly$^{XT}$, and 910 nm from CL51) independently. A difference of ~ 12 % or ~ 36 % was found between Polly$^{XT}$ and CL51 estimated mass concentrations using method #1 or #2, showing that ceilometers are potential tools for mass concentration retrievals with ~ 50 % uncertainty, but with great spatial coverage. The retrieved mass concentration profiles were also compared with MERRA-2 aerosol profiles, where we considered and analysed two scenarios in the SPoI – 1) only smoke particles and 2) mixture of fine dust and smoke aerosols, and reported with the corresponding uncertainties. The inclusion of dust in the retrieved mass concentration is negligible considering the uncertainties; which indicates that ceilometer observations for mass retrievals can be used even without exact knowledge on the composition of the smoke dominant aerosol plume in the troposphere. We demonstrated the potential of the Vaisala CL51 ceilometer to contribute to atmospheric aerosol research in the vertical profile (e.g., to monitor smoke in the troposphere), from sensor-provided attenuated backscatter coefficient to particle mass concentration.

*Data availability.* The data for this paper are available from the authors upon request. Polly$^{XT}$ data quick-looks are available on the PollyNET website (http://polly.tropos.de, last access: 3 May 2021). Ceilometer quick-looks are available on FMI Real-time ceilometer data (http://ceilometer.fmi.fi, last access: 3 May 2021) and data files on request from FMI data archive. Trajectories are calculated with the NOAA (National Oceanic and Atmospheric Administration) HYSPLIT (HYbrid Single-

Particle Lagrangian Integrated Trajectory) model (https://ready.arl.noaa.gov/HYSPLIT.php, last access: 19 March 2021).

Fire data are available at the NASA Worldview application (https://worldview.earthdata.nasa.gov, last access: 2 February 2021). Dust score data from AIRS are available at AIRS application Browse Tool (https://airs.jpl.nasa.gov/map/, last access: 1 July 2021). CALIPSO products are available at the Browse Images Page (https://www-calipso.larc.nasa.gov/products/lidar/browse_images/production, last access: 20 July 2021). MERRA-2 data are available through the NASA Goddard Earth Sciences (GES) Data and Information Services Center (DISC) 550 (https://disc.gsfc.nasa.gov/datasets?project=MERRA-2, last access: 16 April 2021). The AERONET (AErosol RObotic NETwork) data are available at http://aeronet.gsfc.nasa.gov (last access: 3 May 2021).

*Author contributions*. XS analysed the data and wrote the manuscript with contributions from co-authors. XS, TM, An.L, AH, and MK conceptualized the study. EG and AA helped in the validation of lidar and AERONET data analysis. Ar.L 555 provided and assured the quality of the in situ measurements. VB and AD provided and assured the quality of the MERRA-2 data. AK carried out the simulations for the water vapor absorption cross-sections. MK and EO run the lidars and collected the observational data. EO and AH helped in ceilometer data analysis. All authors were involved in the interpretation of the results, reviewing and editing the manuscript.

*Competing interests*. The authors declare that they have no conflict of interest.

*Acknowledgements*. The authors acknowledge the use of imagery from the NASA Worldview application (https://worldview.earthdata.nasa.gov, last access: 3 May 2021), part of the NASA Earth Observing System Data and Information System (EOSDIS). We thank the NOAA Air Resources Laboratory (ARL) for the provision of the HYSPLIT 565 transport and dispersion model used in this publication. The authors acknowledge the MODIS Science, Processing and Data Support Teams for producing and providing MODIS data. The authors acknowledge AIRS research team for the AIRS observation data, and NASA Langley Research Center Atmospheric Sciences Data Center for the data processing and distribution of CALIPSO products. The authors thank Stephanie Bohlmann and Anu-Maija Sundström for fruitful discussions. Tero Mielonen's work was supported by the Academy of Finland (grant No. 308292).

*Financial support*. This research has been supported by the National Emergency Supply Agency (decision number 19078).

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

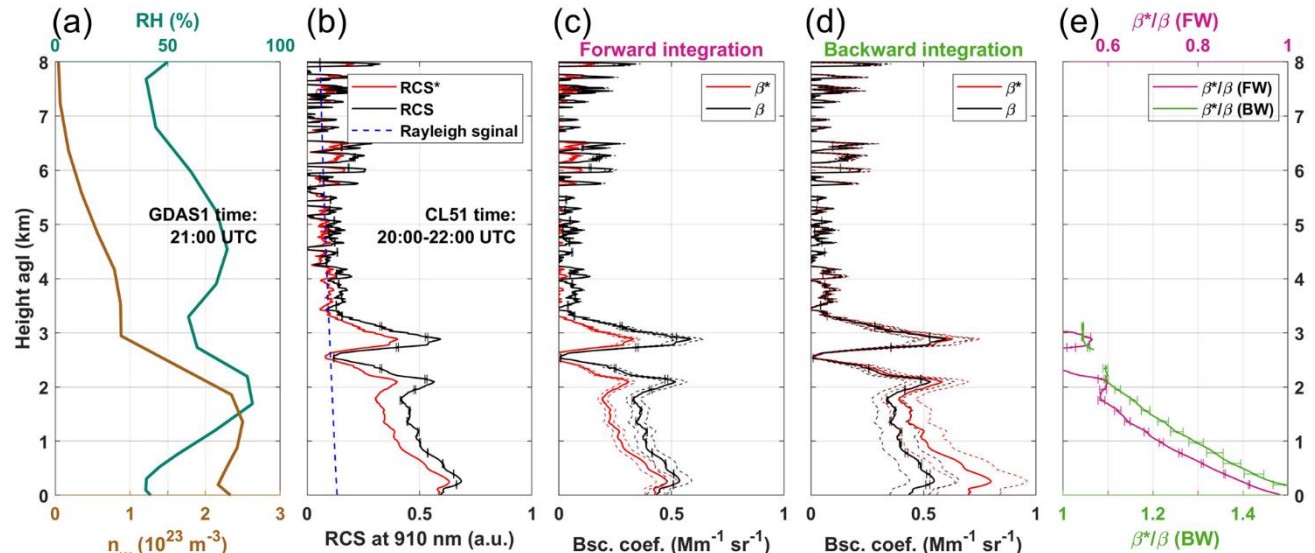

Figure 1. Example of water vapor corrections on 2 h averaged ceilometer data on 5 June 2019 (20:00–22:00 UTC). (a) Relative humidity (RH, teal) and water vapor number density ($n_w$, brown) from GDAS data at 21:00 UTC. (b) Range-corrected signal at 910 nm, without (RCS*, red) or with (RCS, black) water vapor correction, and the hypothetical Rayleigh-signal at 910 nm (dashed blue). (c) Retrieved particle backscatter coefficients: β* without (red) and β with (black) water vapor correction, using forward (FW) integration Klett solution. (d) Same as (c) but application of the backward (BW) integration. (e) Ratio of the retrieved β* and β, when using forward integration (magenta), or backward integration (green). The horizontal lines illustrate the uncertainties range due to wrong assumptions of the central wavelength $\lambda_0 \pm 2$ nm. The uncertainties in backscatter coefficients of the analytical solution were shown by dashed lines.

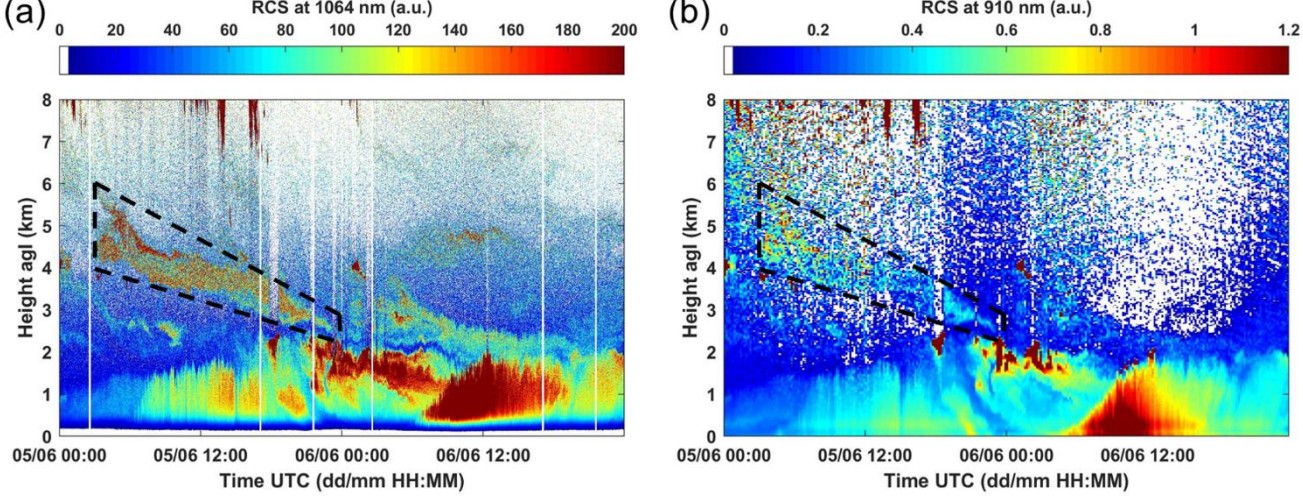

Figure 2. Time–height cross section of range-corrected signal (RCS) (a) at 1064 nm of Polly[XT], and (b) at 910 nm of CL51 ceilometer, on 5 and 6 June 2019 over Vehmasmäki station. Time is given in UTC, and height is above ground. Initial Polly[XT] data were used with temporal and vertical resolutions of 30 s and 7.5 m, respectively. CL51 data were smoothed with temporal and vertical resolutions of 10 min and 70 m, respectively. The SPoI (Smoke Plume of Interest) is inside the black box with dashed lines.

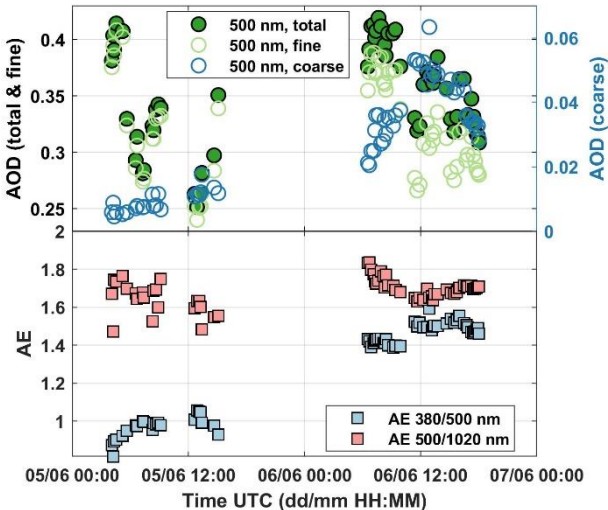

Figure 3. AERONET sun-photometer observations (in Kuopio station, on 5 and 6 June 2019, http://aeronet.gsfc.nasa.gov, last access: 3 May 2021) of (top) 500 nm aerosol optical depth (level 2.0 data) and (bottom) Ångström exponents (AE) computed from the optical depths measured at 380, 500, and 1020 nm. The fine-mode-related (for particle with diameters < 1μm) and coarse-mode-related aerosol optical depth (diameters > 1μm) are shown in addition (top, level 2.0 aerosol spectral deconvolution algorithm (SDA) products).

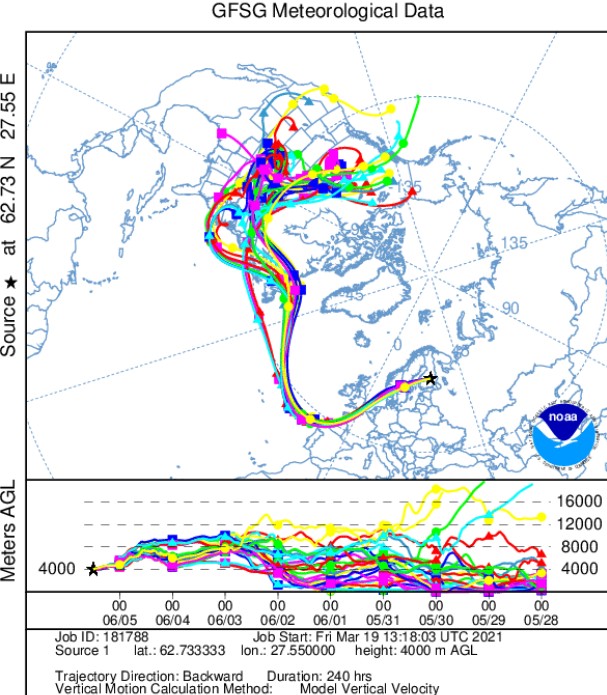

Figure 4. Ten-day backward trajectories from the HYSPLIT model (in ensemble type), ending at 12h UTC on 5 June 2019 for Kuopio, Finland. The end location of the air mass is at 4 km agl in the SPoI (Smoke Plume of Interest).

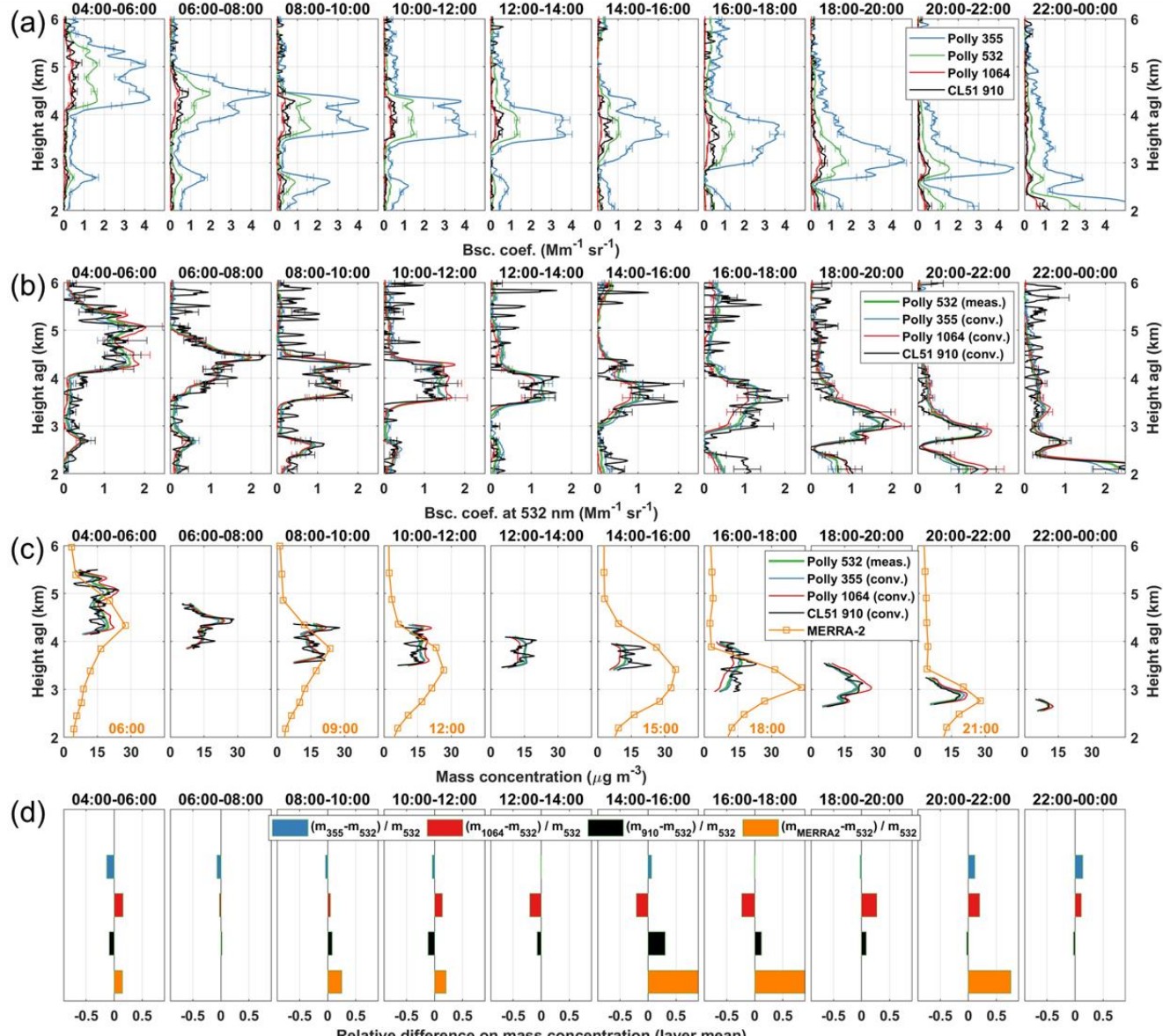

**Figure 5. (a)** Lidar-derived backscatter coefficients (BSC) at 355 (blue), 532 (green), and 1064 nm (red) from Polly[XT], and at 910 nm (black) from CL51. **(b)** BSCs at 532 nm: measured at 532 nm (meas.), or converted (conv.) from measured BSCs at other wavelengths. **(c)** Estimated mass concentration profiles for the SPoI (Smoke Plume of Interest) using BSCs in (b), based on parameters in Table 2-method #1. Mass concentrations from MERRA-2 model are also shown in orange colour with corresponding time given on the bottom right of each panel. **(d)** Relative differences on the mass concentrations (denoted as *m*) estimated from measured/converted BSCs, and of MERRA-2 model, using the one from measured BSC at 532 nm as the reference. 2 h time-averaged lidar profiles are used, with the time slot (UTC) on 5 June 2019 given on top of each panel. The horizontal lines (in a, b) illustrate the uncertainties range. The uncertainties in mass concentrations (in c) are discussed in Sect. 3.2.1.

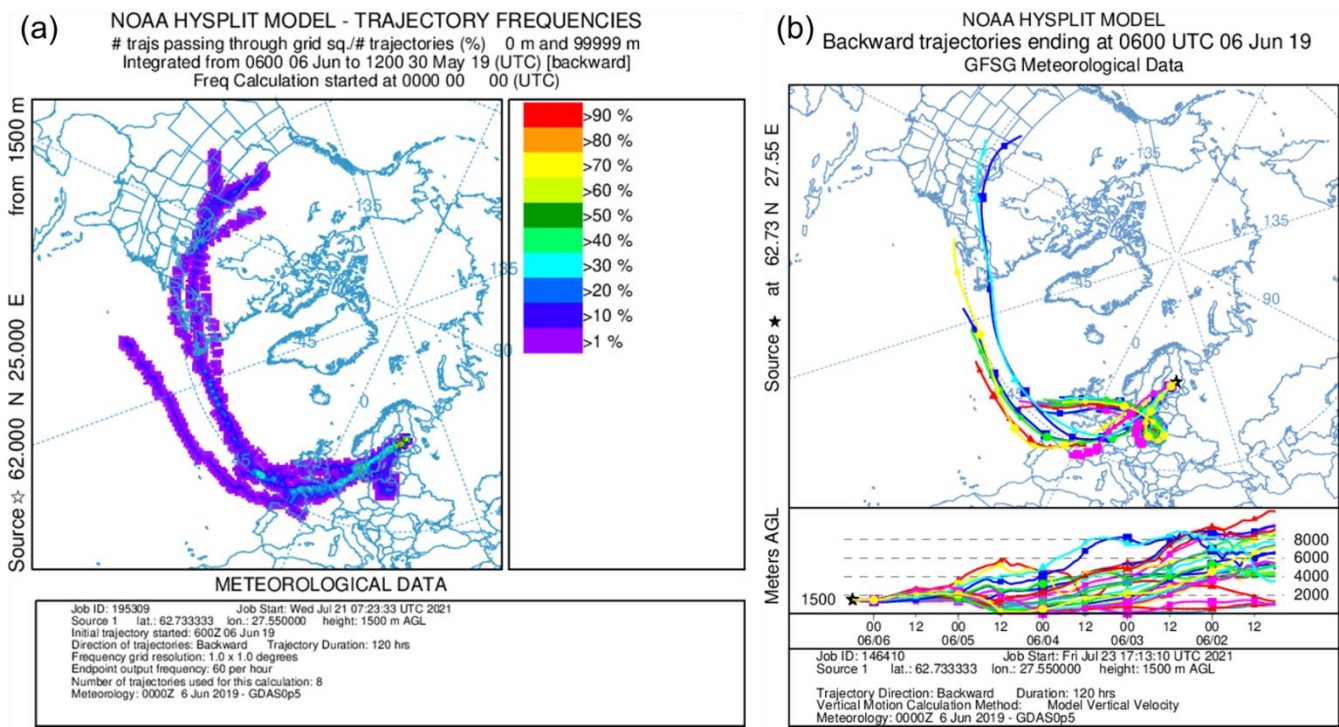

**Figure 6.** Five-day backward trajectories from the HYSPLIT model (a) in Frequency option, and (b) in Ensemble option, ending at 6h UTC on 6 June 2019 for Kuopio, Finland. The end location of the air mass is at 1.5 km agl in the range-transported plume.

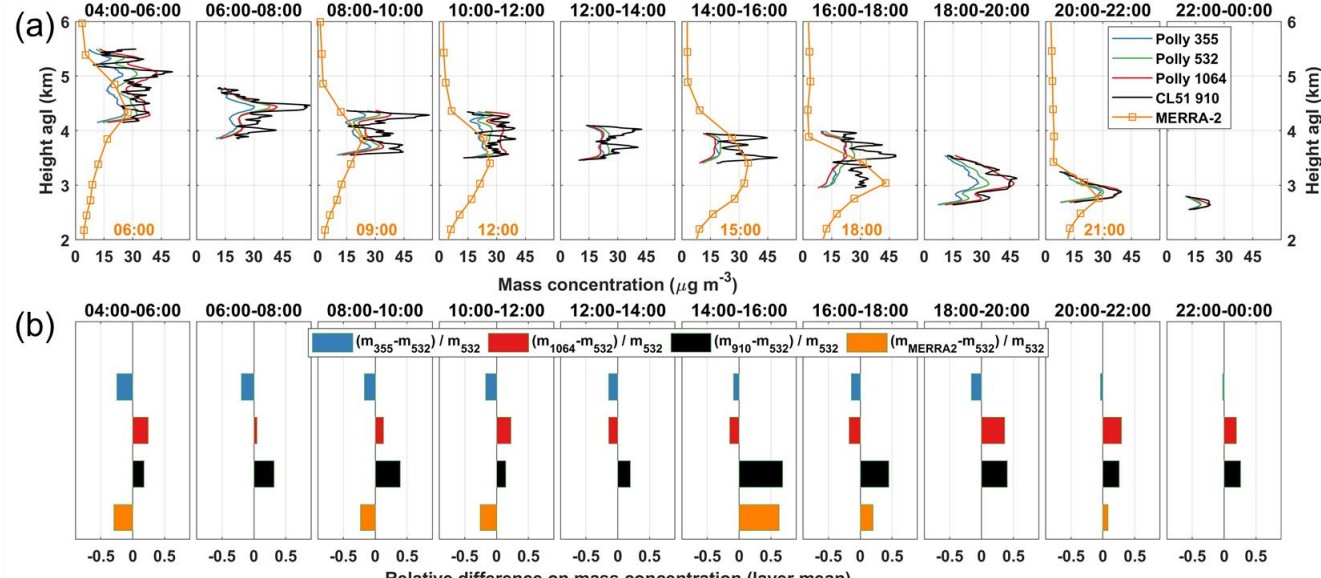

**Figure 7. (a) Estimated mass concentration profiles for the SPoI (Smoke Plume of Interest), based on parameters in Table 2-method #2, using corresponding measured backscatter coefficients (BSCs, Fig. 5 a). Mass concentrations from MERRA-2 model are also shown in orange colour with corresponding time given on the bottom right of each panel. The uncertainties are discussed**

**in Sect. 3.2.2. (b) Relative differences on the mass concentrations (denoted as *m*) estimated from measured BSCs, and of MERRA-2 model, using the one from measured BSC at 532 nm as the reference. 2 h time-averaged lidar profiles are used, with the time slot (UTC) on 5 June 2019 given on top of each panel.**

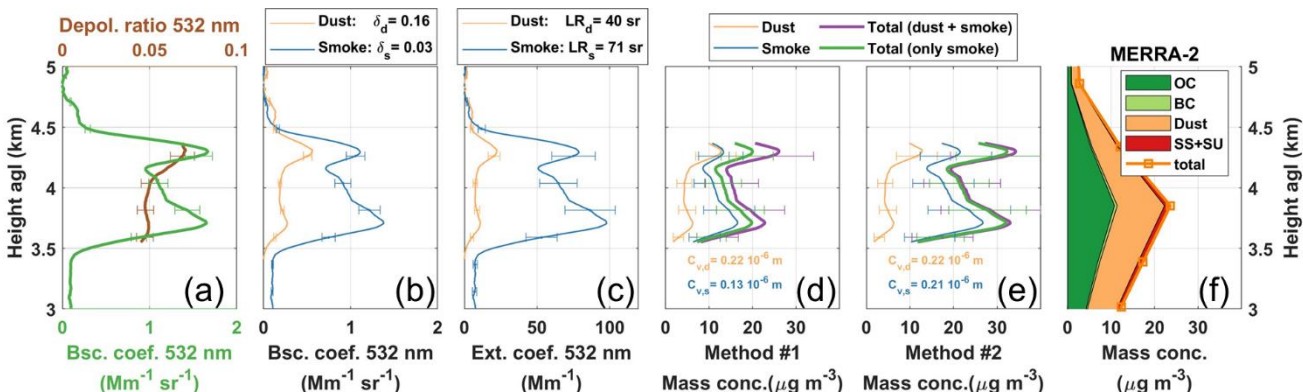

**Figure 8. Lidar products obtained from Polly$^{XT}$ measurements on 5 June 2019, 8–10h UTC (2h signal average). (a) Measured 532 nm total particle backscatter coefficient (green) and particle linear depolarization ratio (brown). (b) Particle backscatter coefficients (BSCs) for fine dust (orange) and smoke (blue) particles, obtained with the POLIPHON method. (c) Respective fine dust and smoke extinction coefficients (EXTs) obtained by multiplying the BSCs (in b) with the lidar ratios. (d,e) The fine dust (orange), smoke (blue) particle mass concentrations derived from the EXT profiles (in c), by using parameters in Table 4 for**

**method #1 (d) and #2 (e). The total mass concentrations of fine dust and smoke mixture (purple), or of only smoke particles (green) are shown. (f) Mass concentrations of organic carbon (OC, dark green), black carbon (BC, light green), dust (orange), sea salt and sulphate (SS+SU, red) from MERRA-2 model, at 9h UTC on 5 June. The total mass concentration profile is also given by orange squares. The horizontal lines illustrate the uncertainties range.**

Table 1. Optical properties (lidar ratio, particle linear depolarization ratios-PDR, backscatter- or extinction-related Ångström exponent-BAE or EAE) of biomass burning aerosols. Layer-mean values of the SPoI (Smoke Plume of Interest) and the standard deviations are given. Optical properties and the effective radius ($R_{eff}$) found in literature of aged forest fire smoke aerosols observed in the troposphere are also given for comparisons. The source regions of these smoke aerosols are all Canada and/or North America.

| | Lidar ratio (sr) | | | PDR (%) | | Ångström exponent | | | $R_{eff}$ (µm) |
|---|---|---|---|---|---|---|---|---|---|
| | 355 | 532 | 1064 | 355 | 532 | EAE 355/532 | BAE 355/532 | BAE 532/1064 | |
| This study | 47 ± 5 | 71 ± 5 | - | 8 ± 2 | 5 ± 1 | 1.4 ± 0.2 | 2.5 ± 0.2 | 2.2 ± 0.3 | - |
| Ancellet et al. (2016) | - | 60 ± 20 | - | - | <5 | - | - | 1.3–2.3 | - |
| Ancellet et al. (2016)[*] | 59 ± 5 | 60 ± 5 | - | 5–8 | 5–10 | - | 2.6 | 1.0–1.3 | - |
| Groß et al. (2013) | - | 69 ± 17 | - | - | 7 ± 2 | - | - | 2.2 ± 0.4 | - |
| Haarig et al. (2018) | 45 ± 5 | 68 ± 9 | 82 ± 27 | 2 ± 4 | 3 ± 2 | 0.9 ± 0.5 | 2.1 ± 0.6 | 0.8 ± 0.3 | 0.17 ± 0.06 |
| Janicka et al. (2017) | 60 ± 20 | 100 ± 30 | - | 1–5 | 2–4 | 0.3–1.7 | 1.7–2.1 | 1.3–1.8 | 0.31–0.36 |
| Müller et al. (2005) | 21–49 | 26–64 | - | - | - | 0.0–1.1 | - | - | 0.24–0.4 |
| Ortiz-Amezcua et al. (2017) | 23–34 | 47–58 | - | - | 2–8 | 0.2–1.0 | 1.2–1.9 | - | 0.21–0.34 |
| Wandinger et al. (2002) and Fiebig et al. (2002) | 40–70 | 40–80 | - | - | 6–11 | - | - | - | 0.27 ± 0.04 |

[*] Biomass burning mixing with a small amount of dust.

Table 2. Parameters required for the mass concentration retrieval using two methods. The smoke mass density and lidar ratio at 532 nm are common parameters required for both methods #1 and #2.

| | Parameter | Wavelength | Value | References |
|---|---|---|---|---|
| Common | Smoke mass density (g cm$^{-3}$) | - | 1.3 | Ansmann et al. (2021) |
| | Lidar ratio (sr) | 532 | 71 ± 5 | This study |
| Method #1 | Smoke volume-to-extinction conversion factor $c_v$ (10$^{-6}$ m) | 532 | 0.13 ± 0.01 | Ansmann et al. (2021) |
| | Backscatter-related Ångström exponent | 355/532 | 2.5 ± 0.2 | This study |
| | | 1064/532 | 2.2 ± 0.3 | |
| | | 910/532 | 1.8 ± 0.2 | |
| Method #2 | Lidar ratio (sr) | 355 | 47 ± 5 | This study |
| | | 1064 | 82 ± 27 | Haarig et al. (2018) |
| | | 910 | 82 ± 27[*] | Haarig et al. (2018) |
| | Fine-mode volume-to-extinction conversion factor $c_v$ (10$^{-6}$ m) | 355 | 0.100 ± 0.002 | This study |
| | | 532 | 0.211 ± 0.003 | (Possible pollution contamination) |
| | | 910 | 0.620 ± 0.002 | |
| | | 1064 | 0.902 ± 0.004 | |

[*] LR values measured at 1064 nm are used for LR at 910 nm.

Table 3. Relative uncertainties in the input parameters and in the retrieved products (in bold). The uncertainty origins are given for input parameters and denoted as: R-Raman measurement available, E-only elastic measurement for the retrieval, L-literature, A-assumption. The uncertainty in the smoke mass density ($\rho$) was assumed as 20 % as in Ansmann et al. (2021). Different retrieval information (R or E) is available at each wavelength with a different system (Polly^XT or CL51), thus different uncertainties in the backscatter coefficients ($\beta$) and lidar ratio ($LR$) are considered. The uncertainty in the smoke volume-to-extinction conversion factor ($c_v$) was assumed as 10 % for both methods, as given in Ansmann et al. (2021). The relative uncertainties in the mass concentration ($m$), backscatter-related Ångstöm exponent ($BAE$), and converted backscatter coefficient ($\beta_{conv}^{532}$) are obtained by the error propagation applied to Eqs. 1-5.

|  |  | Polly^XT | | | CL51 |
|---|---|---|---|---|---|
|  | $\lambda$ (nm) | 532 | 355 | 1064 | 910 |
|  | Uncertainty |  |  |  |  |
| Common | $\Delta\rho/\rho$ | 0.20 (L) | | | |
|  | $\Delta\beta/\beta$ | 0.10 (R) | 0.10 (R) | 0.15 (E) | 0.20 (E) |
| Method #1 | $\Delta c_v/c_v$ | 0.10 (L) | - | - | - |
| if $\beta(532)$ | $\Delta LR/LR$ | 0.20 (R) | - | - | - |
| available | $\Delta BAE/BAE$ * | - | **0.14** | **0.12** | **0.24** |
|  | $\Delta\beta_{conv}^{532}/\beta_{conv}^{532}$ ** | - | **0.18** | **0.24** | **0.31** |
|  | $\Delta m/m$ | **0.32** | **0.36** | **0.40** | **0.45** |
| if $\beta(532)$ | $\Delta LR/LR$ | 0.30 (A) | - | - | - |
| not available | $\Delta BAE/BAE$ * | - | 0.30 (A) | 0.30 (A) | 0.30 (A) |
|  | $\Delta\beta_{conv}^{532}/\beta_{conv}^{532}$ ** | - | **0.33** | **0.51** | **0.36** |
|  | $\Delta m/m$ | - | **0.67** | **0.52** | **0.54** |
| Method #2 | $\Delta c_v/c_v$ | 0.10 (A) | 0.10 (A) | 0.10 (A) | 0.10 (A) |
|  | $\Delta LR/LR$ | 0.20 (R) | 0.20 (R) | 0.30 (L) | 0.40 (A) |
|  | $\Delta m/m$ | **0.32** | **0.32** | **0.42** | **0.52** |

* Wavelength pair of $\lambda$ and 532, ** Converted backscatter coefficient at 532 nm from $\lambda$.

Table 4. Parameters required for the mass concentration retrieval, considering fine dust and smoke mixture.

|  |  | Smoke | | Fine dust | |
|---|---|---|---|---|---|
| Mass density (g cm^-3) | | 1.3 | (Ansmann et al., 2021) | 2.6 | (Ansmann et al., 2012) |
| Depolarization ratio at 532 nm | | 0.03 | (Haarig et al., 2018) | 0.16 | (Sakai et al., 2010) |
| Lidar ratio at 532 nm (sr) | | 71 | this study | 40 | (Ansmann et al., 2019) |
| Volume-to-extinction conversion | method #1 | 0.13 | (Ansmann et al., 2021) | 0.22 | (Ansmann et al., 2019) |
| factor $c_v$(532 nm) (10^-6 m) | method #2 | 0.21 | this study | 0.22 | (Ansmann et al., 2019) |