# Peer review of "Mass concentration estimates of long-range-transported Canadian biomass burning aerosols from a multi-wavelength Raman polarization lidar and a ceilometer in Finland"

_Atmospheric Measurement Techniques, 2021_

## Referee Comment (RC3)

General

The paper contains smoke observations over Finland. But this aspect alone is, to my opinion, not sufficient to justify publication. Meanwhile there are so many smoke observations with lidar in the literature (see review of Adam et al., 2020) and even over the North Pole (Ohneiser et al., 2021). Therefore, the goals of the paper need to better emphasized: lidar-ceilometer observations and comparison with model results is probably one goal. Another goal is the careful analysis (some kind of a feasibility study) to what extent Vaisala ceilometers (and these huge ceilometer networks) can contribute to tropospheric smoke monitoring (even in terms of mass concentration profiling). The paper is worthwhile to be published, however only after significant improvement. Furthermore, the paper contains many speculative and questionable aspects. Their own AERONET approach to derive smoke conversion factors is unacceptable. So, there are many parts that need to be significantly improved.

Major revisions are required.

Details:

Abstract:

It should be clearly stated in the beginning: What is the main goal of the paper, what is new in this paper (in view of the numerous smoke observations with lidar in the literature, see review of Adam et al., 2020). First lidar smoke observations over Finland …. is not a convincing argument (or goal). Recently, TROPOS people even measured smoke over the North Pole (Engelmann et al., 2020, Ohneiser et al., 2021) … with lidar aboard an ice breaker.

To my opinion, to combine lidar and ceilometer observations (and even to include modelling) is an attractive approach. And especially, if the main goal is: … to demonstrate the usefulness of a Vaisala ceilometer to monitor smoke in the troposphere!

However, feel free to define your specific goals! This is just a suggestion. In this context, you can then easily present all your nice smoke results on changing depolarization ratios, on this unique smoke feature with larger lidar ratios at 532 than at 355 nm, and the comparison with model results for smoke.

Introduction:

P 2-3: The Introduction should be improved. You mention the Mueller1999 paper, then I would add the Mueller2005 paper as well because that paper is directly related to smoke observations (and lidar inversion application). Furthermore, you need to mention this Adam 2020 review paper!

In the next step, you may want to continue with network activities (before you introduce the ceilometer network aspect), and maybe, also CALIPSO observations. Again, with clear focus on smoke. There are these Baars2019 and Khaykin2018 papers as examples for network and space lidar activities. This would show the added value towards regional to global scale smoke characterization when using networks. This motivates, to my opinion, then the next ste0: …. to analyse to what extent the existing and exciting (European) ceilometer infrastructure could do in case of smoke monitoring… and so on…. All this would corroborate the importance of the paper. Are there some smoke observations with ceilometers

in the literature (I am not sure)? If yes, should ber cited. If not, that would be new point to be mentioned! One may also indicate similar approaches such as the ceilometer observations of volcanic aerosols (Eyjafjalla volcanic aerosol, Emeis and Flentje papers in 2010/2011?) to indicate the usefulness of modern ceilometers to detect aerosols (and not only clouds).

Feel free to define your own specific goals of the paper. It is not very clear to me at the moment what the goals are.

Now some more detailed remarks:

P6, line 176: I do not believe that you can get the backscatter coefficient at 910 or 1064 nm with an uncertainty of less than 10%. The uncertainty in the reference value is too large. And a proper Rayleigh fit at these long wavelengths almost impossible. The uncertainty is certainly always in the range of 20-30% at 910 or 1064 nm for the backscatter coefficient. And the conversion (backscatter to extinction) will introduce another 20-40% uncertainty in the case of smoke layers. The lidar ratio for smoke was found to be 50, 60, 70, 80, even 110 sr in smoke observation (see Adam et al, including the ACPD version and supplementary tables). So, using, e.g., 75 sr as smoke lidar ratio at 532 nm, and the range is from 50 to 100 sr, than the error is 33%. The uncertainties are probably similar for 910 nm.

P7, lines 201 – 214: I speculate that there was an air mass transport from central Europe to Finland at heights below 3 km height (not presented), when I see the backward trajectory figure for the arrival height of 4 km. And this aged European haze widely determined the observed AOD over the field sites. You mention 500 nm AODs of 0.24-0.42 (as written on page 6). And for the smoke layer the 532 nm AOD was found to be 0.02 to 0.13 (page 7, Sect 3.1.). So the smoke impact was at least not dominating. This means that the AERONET observations cannot be used to derive smoke conversion factors. This point will be further discussed below.

To continue: Surface (in situ) observation cannot be used when discussing lofted layers. And the in situ measured aerosol values are most probably enhanced because of the advected central European haze. So, the final paragraph in Sect. 3 (before Sect. 3.1) makes no sense, and should be skipped.

P8, line 234: Sedimentation of large particles is not a good argument here. Smoke particles always show a pronounced accumulation mode, so difference in falling speed is low, when coarse mode particles are absent. Particle aging is more likely. Smoke aging process mainly occur in the first 36-48 hours after emission, and afterwards aging is slow. At the end of this aging process, the particles are usually spherical or almost spherical in shape. Particles show an almost perfect core-shell structure (coating, OC material) and the shell is often liquid at lower heights. And the probability that smoke particle are glassy (not perfectly round) increase with decreasing temperature. That could also be a reason that you saw a decreasing trend in the depolarization values with decreasing height.

P9, line 256: It makes no sense to me at all to use the actual AERONET data to derive smoke conversion factors. As mentioned, the AOD was obviously dominated by European pollution, so that the conversion factors reflect European fine mode haze properties. All the efforts to get proper conversion factors from AERONET (dust, smoke, marine, etc.) were done in regions with pure dust or marine or smoke conditions, etc. One should therefore use the conversion factors presented in this Ansmann 2020 paper, or you try to use the Polly multiwavelength information (inversion) to obtain the smoke volume concentration in the

smoke layer together with the backscatter and extinction coefficients in these layers, and in this way the required smoke conversion parameters. Your conversion factor of 0.21 perfectly describes the conversion factor for urban haze. The smoke conversion factors are in the range from 0.12-0.15, and thus considerably lower.

P10, lines 288-289, please re-calculate the uncertainties by assuming 20% (BSC), 30% (LR), 20% (conversion factor from literature) and 20% (particle density), probably the uncertainty is 40-50%.

P10, lines 295-300, this is a 'pure' speculation about dust (only fine-mode dust, no coarse mode dust), is my feeling. On the other hand, the enhanced depolarization ratio can easily be explained by non-spherical smoke. Already small deviations from the ideal spherical shape causes depolarization as Gialitaki etal., ACP, 2020 shows.

P10, lines 301-318, These paragraphs do not make any sense. I would remove this part. It is pure speculation. Sure, you may have fine dust, but without the presence of any coarse dust? Is that possible? And again, non-spherical smoke is a convincing argument for the enhanced depol values.

Figure 6: If you include a 40 or 50% uncertainty bar to the mass concentration values, you do not need to speculate about any dust contribution!

In general, I miss uncertainty bars in Figures 6 and 7. Not many, but at least one or two per lidar and ceilometer profile!

P11, L319-334. All this should be removed, just speculation, simply not convincing! Impossible, to accept that as a reviewer!

Now we need conclusions: One conclusion should deal with the question: What is now the value of the ceilometer? The ceilometer is able to detect smoke layers even in the middle to upper troposphere? With what overall uncertainty? What about mass retrieval from ceilometer observations? Possible? Yes or no? Conversion factors for 910 nm are not available. How to proceed? ..with 910/532 nm smoke backscatter color ratios? ..to convert 910 backscatter into 532 nm backscatter for which smoke conversion factors are available.

---

## Author Response (AR1)

**Response to Referee #1**

Thank you for carefully reading the manuscript and providing useful suggestions to improve the paper. The replies to the referee comments are given below. The referee comments are highlighted in blue with our responses in black. Some comments concerning similar issues are grouped together. The sentences in the manuscript are between the quotation marks, with the modifications in the revised manuscript in red.

**The manuscript presents a Canadian biomass burning event measured by a multiwavelength Raman lidar (PollyXT) and a Vaisala CL51 ceilometer in Finland. The aerosol backscatter coefficients are converted to smoke mass concentration following the methodologies in literature. Comparison with model from MERRA-2 are shown as well.**

**I suggest the publication of this manuscript after addressing all the points raised by reviewers.**

**Please see below some suggestions and comments:**

**Pp 6, l 163-165: comment on the uncertainty of the water vapor absorption profiles used for the correction in the ceilometer backscatter profiles**

Thank you for the comments. We have added descriptions about the water vapor corrections in section 2.3 in the revised version:
"

Wiegner and Gasteiger (2015) state that the annual variability of pressure and temperature has no significant influence on the water vapor absorption cross-sections. It is possible to use the tabulated mean absorption cross-section to calculate an approximative water vapor transmission with a high accuracy (the inherent error of the squared water vapor transmissions is <0.3 %; more details are given in section 4 in Wiegner and Gasteiger, 2015).

"

**Fig. 1 Please add uncertainties to profiles. Also, mention the method you use to compute it.**

We have added the uncertainty related to the wrong assumptions of the central wavelength when applying the water vapor correction in Fig.1. We also added the uncertainties in backscatter coefficients of the analytical solution in Fig.1. Further, we have added more description on the uncertainty study about forward and backward method in the revise version.
"

The uncertainties range due to wrong assumptions of $\lambda_0 \pm 2$ nm is given by the horizontal lines in Fig. 1. The uncertainties in backscatter coefficients of the analytical solution were also shown by dashed lines. As the water vapor contribution cannot be neglected at Kuopio during summer, the water vapor corrections have been applied to CL51 data in this study.

The retrieval methods for deriving the backscatter coefficient from ceilometers are quite mature (Wiegner et al., 2014; Wiegner and Geiß, 2012). Under favourable conditions, a relative error of the backscatter coefficient on the order of 10 % seems feasible with a careful calibration by applying the forward integration. On the contrary, significant temporal averaging of ceilometer data is required for performing a Rayleigh calibration, as the detection of molecular signals is intrinsically very difficult. Binietoglou et al. (2011) propose a two-step approach, resulting promising agreement comparing to their lidar PEARL (Potenza EARLINET Raman lidar). The uncertainty of the backscatter coefficient could be in the range of 20–30 % using the backward integration. The advantage of the forward algorithm is that calibration is required only occasionally, and it is not affected by the low SNR in the upper troposphere. However, the accuracy in deriving extinction coefficients is limited due to the unknown LR at 910 or 1064 nm and its uncertainties. In particular the presence of multi-layered aerosol distributions (with different aerosol types) may introduce more uncertainties. In addition, the uncertainty due to the neglecting the water vapor increased with the distance from the chosen reference height. In this study, we applied the Klett method (Wiegner et al., 2014) by defining the reference height as close as to the layer of interest, so that the error propagation (due to uncertainties of LR and water vapor transmission) would be minimized for that layer.

[Figure]

**Figure 1.** Example of water vapor corrections on 2 h averaged ceilometer data on 5 June 2019 (20:00–22:00 UTC). (a) Relative humidity (RH, teal) and water vapor number density ($n_w$, brown) from GDAS1 data at 21:00 UTC. (b) Range-corrected signal at 910 nm, without (RCS*, red) or with (RCS, black) water vapor correction, and the hypothetical Rayleigh-signal at 910 nm (dashed blue). (c) Retrieved particle backscatter coefficients: $\beta^*$ without (red) and $\beta$ with (black) water vapor correction, using forward (FW) integration Klett solution. (d) Same as (c) but application of the backward (BW) integration. (e) Ratio of the retrieved $\beta^*$ and $\beta$, when using forward integration (magenta), or backward integration (green). **The horizontal lines illustrate the uncertainties range due to wrong assumptions of the central wavelength $\lambda_0 \pm 2$ nm. The uncertainties in backscatter coefficients of the analytical solution were shown by dashed lines.**

"

**Pp 6, l 181: please add uncertainty for LR**

We have added the uncertainty in the revised version as:
"
A value of 82 sr for LR, as measured at 1064 nm (82 ± 22 sr in Haarig et al., 2018), was assumed as being appropriate for use at 910 nm in this study.
"

**Pp 6, l 192: please comment on the existence of the pollen. How do you know is pollen? Did you measure / estimate it? I guess it is typical to find pollen in June.**

Thank you for the suggestion. We had in situ pollen measurements during the period. And you are right, June is a typical time of pine pollen for our site. This information was added in section 3 of the revised version:
"
Our in situ pollen measurements (more information about pollen instruments can be found in Bohlmann et al., 2021) shows high pine pollen loading (highest 2 h pollen concentrations were ~ 3000 $m^{-3}$ on 5 June and ~ 7000 $m^{-3}$ on 6 June). Although also of interest, the analysis of the pollen layer is out of the scope of this paper.
"

**Pp 7, l 219-220: spatial resolution remains at 7.5m? Later you mention 11 bins sliding average for lidar and 7 bins for ceilometer. Please clarify.**

**Pp 9, l 274-275. For 11 bins gliding average over lidar profiles you obtain 82.5 m effective resolution. For the ceilometer, you obtain 70 m resolution. I was expecting more smoothing over ceilometer as it is much noisier. Please comment your choices.**

Thank you for the suggestion. We have clarified this in the revised version.
In section 2.2 (Polly$^{XT}$ lidar) we added:

"

The initial spatial and time resolution is 7.5 m and 30 s, respectively. The laser beams are tilted to an off-zenith angle of 5° to avoid specular reflections from horizontally aligned ice crystals. For the calculation of optical properties in this study, the profiles were temporally averaged in 2 h intervals, and smoothed with a vertical gliding averaging window length of 11 bins (a vertical range of ~ 82 m).

"

In section 2.3 (Ceilometer and data processing) we added:

"

The signal-to-noise ratio (SNR) for raw CL51 backscatter signals above the boundary layers is weak, hence some temporal averaging and vertical smoothing were required when performing further analysis. In this study, CL51 signals were smoothed with a vertical gliding averaging window length of 7 bins (a vertical range of ~ 70 m). The profiles were temporally averaged in 10 min intervals for the time–height cross section quick look, and in 2 h intervals to calculate the optical properties.

"

We have also recalled this information in section 3.1 (Optical properties) for the clarity:

"

Layer-mean values of optical properties of the SPoI were derived and are given in Table . Two-hour time-averaged, and vertical smoothed (with a smoothing window of ~ 82 m) lidar profiles were used in order to increase SNR.

"

In our data processing program, we used odd number of bins as the sliding window, and we select 11 bins for PollyXT and 7 bins for ceilometer, so that they are more or less the similar vertical smoothing (we can also choose 9 bins for ceilometer). As we mainly consider the layer mean values, the vertical smoothing doesn't have significant impacts on the results. We think 7 bins are good enough for our smoke layer study, as the CL51 has enough power.

**Pp 7, l 195-199: please comment on the choices for AE. Why did you choose the ratios 500/870 and 380/500? Is it the later chosen for comparison with lidar's EAE (e.g. Muller et al., 2013; Nicolae et al., 2019). When you refer to fine particles do you refer to those smaller than 1 um?**

**Pp 8, l 227-231: EAE of 1.4 is suggested as a delimitation from fresh and aged smoke along with LR532 > LR355 (Nicolae et al., 2013). Higher values of EAE correspond to smaller effective radius (e.g., Muller et al., 2005). Please clarify how you consider the range of fine particles. If you consider fine particles those smaller than 1 um (as for photometer), then we measure fine mode particles with the lidar most of the time. On the other hand, one can consider 500 nm as the delimitation between fine-mode and coarse-mode (e.g., Muller et al. 2016). Mamouri and Ansmann refer to fine dust if the particle's radius is < 500 nm. Please comment on the value of EAE derived from lidar (1.4) and discuss the relationship with AE from photometer for 380/500 assuming the value is similar with that corresponding to 355/532. As seen in Fig. 3, AE for 5th of June is around 0.8-1.05. I would have expected closer values for EAE and AE.**

Thank you for the comment. We have changed this figure from AE 500/870 to AE 500/1020, as follows. These selected 3 wavelengths (380, 500, and 1020) in the revised version are more close to our PollyXT lidar wavelengths (355, 532, and 1064). We found that the values of AE 500/870 and AE 500/1020 were quite similar. Here, we used the AEROENT level 2.0 aerosol spectral deconvolution algorithm (SDA) products, which yield fine (sub-micron) and coarse (super-micron) aerosol optical depths at a standard wavelength of 500 nm. Related information has been added in the revised version.

[Figure]

**Figure 3.** AERONET sun-photometer observations (in Kuopio station, on 5 and 6 June 2019, http://aeronet.gsfc.nasa.gov, last access: 3 May 2021) of (top) 500 nm aerosol optical depth (level 2.0 data) and (bottom) Ångström exponents (AE) computed from the optical depths measured at 380, 500, and 1020 nm. The fine-mode-related (for particle with diameters < 1μm) and coarse-mode-related aerosol optical depth (diameters > 1μm) are shown in addition (top, level 2.0 aerosol spectral deconvolution algorithm (SDA) products).

We can compare these sun-photometer AE with lidar's EAE. However, note that the sun-photometer AE is for the total atmospheric column (as we mentioned in section 2.1), whereas lidar's EAE in Table 1 is only for the smoke layer. On 5th, there were cirrus presence (this is also the reason why there was no AERONET inversion product available on 5th) above the smoke layer, and pollen or other aerosols presence in the PBL; please find following figure of PollyXT RCS at 1064 nm (in the manuscript, the y-axis of Fig.2a was cut at 8 km). Also, the lidar's EAE was only available for night-time (between 5[th] and 6[th]), but sun-photometer AE was for daytime. If we interpolate the sun-photometer AE of 5[th] and 6[th], a better agreement can be found for lidar's EAE. We have added descriptions for the clarity in the revised version.

[Figure]

Thank you for the suggestion about "EAE of 1.4 is suggested as a delimitation from fresh and aged smoke", we have added such information in the revise version.

We haven't performed the microphysical analysis to estimate the effective radius, but we have added descriptions about possible effective radius value (0.23 μm), based on figure 6 in Muller et al. 2005. We have also added one column of the effective radius values from literature in Table 1 (please check our reply to your later comment).

Please see our modifications in the revised version about previous points as follows:
In section 3.1:
"

Nonetheless, the wavelength dependence of the extinction coefficient for the 355–532 nm spectral range is much weaker, with an extinction-related Ångström exponent (EAE) of ~ 1.4. Nicolae et al. (2013) state that the EAE can be used for identifying the evolution of ageing processes of biomass burning aerosol, as it decreased from 2 for fresh to ~ 1.4–0.5 for ages biomass burning aerosols. The microphysical analysis was not performed in this study; yet the measured EAE would be related to the effective radius of ~ 0.23 μm, when considering the relationship between EAE and effective radius of forest fires smoke reported by Müller et al., (2005) (c.f., Fig. 6 in that paper). This estimated effective radius value is consistent with those of aged smoke aerosols reported in literature (Table 1). The AERONET sun-photometer Ångström exponent at 380–500 nm on 5 June showed lower values than lidar EAE at 355–532 nm; possible cirrus contamination could partly explain as sun-photometer data are for the total atmospheric column. Note that lidar's EAE was only available for night-time (between 5th and 6th).

"

[Figure]

**Figure 6.** Correlation plot of (a) Ångström exponent versus effective radius and (b) single-scattering albedo versus imaginary part of the complex refractive index. Open circles denote cases of forest fire smoke. Solid circles describe the case of anthropogenic pollution. Regression lines denote best linear fit. In the case of Figure 6b, fit is shown under consideration of all data points (solid line) and if only data points representing forest fire smoke are considered (dash-dotted line).

Figure 6 from Muller et al. 2005.

Regarding the range of fine particles, we consider fine particles as for the photometer. Thank you for pointing it out, we made modifications to clarify it in the revised version.
In section 2.1 for AERONET data descriptions, we added:
"

The volume particle size distribution was retrieved in the range of radius of 0.05–15 μm; the minimum within the size interval from 0.439–0.992 μm was used as a separation point between fine and coarse mode particles.

"
in the new section 3.2.2 when we used AERONET fine mode data, we added:
"

From the size distribution, the separation points between fine and coarse mode particles were found as ~ 0.576 μm (the size classes 1–10 were considered for fine-mode aerosols).

"
in the new section 3.3 when we applied the separation method, we added:
"

The POLIPHON (Polarization lidar photometer networking) method (Mamouri and Ansmann, 2014, 2017) was applied to separate fine dust (particles with radius < 500 nm) and biomass burning aerosols for the SPoI.

"

**Pp 8, l 241-243: PDR@532 > 5% are observed in literature for aged smoke. Please check the literature review by Adam et al. 2020 (see Supplement).**

**Table 1. You can add values for BAE from Ancellet by converting CR to BAE. Also, you can add the case of pure BB. More values from literature are given by Adam et al. 2020, Supplement.**

Thank you for the suggestion. We have enriched the information for Table 1. We have removed the case from Muller et al. 2007, as the source region was Siberia, but added other references (with source regions for Canada or North America) as shown in red in the table. We have also added one column of the effective radius values.
We have converted CR values in Ancellet et al. 2016 to BAE as shown in table below.
"
**Table 1. Optical properties (lidar ratio, particle linear depolarization ratios-PDR, backscatter- or extinction-related Ångström exponent-BAE or EAE) of biomass burning aerosols. Layer-mean values of the SPoI (Smoke Plume of Interest) and the standard deviations are given. Optical properties and the effective radius ($R_{eff}$) found in literature of aged forest fire smoke aerosols observed in the troposphere are also given for comparisons. The source regions of these smoke aerosols are all Canada and/or North America .**

| | Lidar ratio (sr) | | | PDR (%) | | Ångström exponent | | | $R_{eff}$ (µm) |
|---|---|---|---|---|---|---|---|---|---|
| | 355 | 532 | 1064 | 355 | 532 | EAE 355/532 | BAE 355/532 | BAE 532/1064 | |
| This study | 47 ± 5 | 71 ± 5 | - | 8 ± 2 | 5 ± 1 | 1.4 ± 0.2 | 2.5 ± 0.2 | 2.2 ± 0.3 | - |
| Ancellet et al. (2016) | - | 60 ± 20 | - | - | <5 | - | - | 1.3–2.3 | - |
| Ancellet et al. (2016)[*] | 59 ± 5 | 60 ± 5 | - | 5–8 | 5–10 | - | 2.6 | 1.0–1.3 | - |
| Gross et al. (2013) | - | 69 ± 17 | - | - | 7 ± 2 | - | - | 2.2 ± 0.4 | - |
| Haarig et al. (2018) | 46 ± 6 | 67 ± 4 | 82 ± 22 | 2 ± 4 | 3 ± 2 | 0.9 ± 0.5 | 2.1 ± 0.6 | 0.8 ± 0.3 | 0.17 ± 0.06 |
| Janicka et al. (2017) | 60 ± 20 | 100 ± 30 | - | 1–5 | 2–4 | 0.3–1.7 | 1.7–2.1 | 1.3–1.8 | 0.31–0.36 |
| Muller et al. (2005) | 21–49 | 26–64 | - | - | - | 0.0–1.1 | - | - | 0.24–0.4 |
| Ortiz-Amezcua et al. (2017) | 23–34 | 47–58 | - | - | 2–8 | 0.2–1.0 | 1.2–1.9 | - | 0.21–0.34 |
| Wandinger et al. (2002) and Fiebig et al. (2002) | 40–70 | 40–80 | - | - | 6–11 | - | - | - | 0.27 ± 0.04 |

[*] Biomass burning mixing with a small amount of dust.
"

We agree that the conclusion of "PDR were slightly higher than the values given in the literature" is too speculative, we have removed related sentences and made modifications in the revised version:
"
The smoke particles caused slightly enhanced particle linear depolarization ratios (PDR) at 355 nm (532 nm) with a mean value of 0.08 ± 0.02 (0.05 ± 0.01) in the smoke layer, suggesting the presence of partly coated soot particles or particles that have mixed with a small amount of dust or other non-spherical aerosol type. The layer-mean PDR at 355 nm (532 nm) decreased during the day, from ~ 0.11 (0.06) in the morning to ~ 0.05 (0.04) in the evening. The decrease of the PDR with time could be linked to the particle aging and related changed in the smoke particle shape properties, as stated by Baars et al. (2019). The relative humidity (RH) profiles from GDAS1 data showed low values in the lower atmosphere (<60 % below 6 km) before 15h UTC, and even lower RH (<40 %) at the SPoI altitude. RH slightly increased in the evening. The signal in the 407 nm Raman-shifted channel was used to determine the water vapor mixing ratio profile during night-time, showing that the layer-mean RH changed from ~ 27 % at 19 h to ~ 38 % at 23 h, which was associated with the advection of a moister air mass with a water vapor mixing ratio close to 1–3 g kg[-1]. The smoke particles were dry, and then captured water vapor in the atmosphere during the evening. The decreasing temperature and increasing RH also increase the probability that smoke particles become glassy. The depolarization ratios of aged biomass burning aerosols (originating from Canada and/or North America) reported in the literature (Table 1) range from 0.01 to 0.11 (0.01 to 0.08) at 532 nm (355 nm). More information of the aged smoke from other regions can be found in the literature review by Adam et al. (2020, see the Supplement).
"

**Pp 9, l 278. You mention a maximum value of 45 ug/m3. Please mention to which profile you refer to (355, 532, 910) and the time interval. Also, please add comments on morning and night values as you mention in abstract and summary. Please add the profile at 1064nm as this one compares better with 910nm (closer wavelengths).**

**Pp 11, l 319: please add numerical values when discussing 'large discrepancies' or good agreements between mass concentrations estimated from 355, 532 and 910. A simple way is to compare the mean value in the layer from each profile. Then you can mention the minimum and maximum differences between profiles.**

**Pp 12, 355: as said, add few comments in the text about those 30 and 20 ug/m3 values in the morning and in night. Nothing is specified in the main text. Mention the time it was observed and the profiles (e.g., 355, 532, 910). When looking at Fig. 6, I can see values of mass concentration around 30 ug/m3 in 532 profile at 06:00 and 09:00. However, I can see also values at 30 ug/m3 at 21:00. I don't know where 20 ug/m3 is observed.**

**Pp 12, l 354: please comment quantitatively on 'good agreement'; see above. Also, comment on the agreement between ceilometer and the retrieval of mass concentration starting from 1064 backscatter profile (first, add this profile).**

**Fig. 6. Please add uncertainties to profiles.**

Thank you for the very useful suggestions.

We have added the 1064 nm profiles for additional information. The factor at 1064 nm were added in Table 2 in the revised version:
"

**Table 2. Parameters required for the mass concentration retrieval using two methods. The smoke mass density ($\rho$) and lidar ratio at 532 nm are common parameters required for both methods #1 and #2.**

| | Parameter | Wavelength | Value | References |
|---|---|---|---|---|
| Common | Smoke mass density (g cm$^{-3}$) | - | 1.3 | Ansmann et al. (2021) |
| | Lidar ratio (sr) | 532 | $71 \pm 5$ | This study |
| Method #1 | Smoke volume-to-extinction conversion factor $c_v$ ($10^{-6}$ m) | 532 | $0.13 \pm 0.01$ | Ansmann et al. (2021) |
| | Backscatter-related Ångström exponent | 355/532 | $2.5 \pm 0.2$ | This study |
| | | 1064/532 | $2.2 \pm 0.3$ | |
| | | 910/532 | $1.8 \pm 0.2$ | |
| Method #2 | Lidar ratio (sr) | 355 | $47 \pm 5$ | This study |
| | | 1064 | $82 \pm 22$ | Haarig et al. (2018) |
| | | 910 | $82 \pm 22^{*}$ | Haarig et al. (2018) |
| | Fine-mode volume-to-extinction conversion factor $c_v$ ($10^{-6}$ m) | 355 | $0.100 \pm 0.002$ | This study |
| | | 532 | $0.211 \pm 0.003$ | (Possible pollution contamination) |
| | | 910 | $0.620 \pm 0.002$ | |
| | | 1064 | $0.902 \pm 0.004$ | |

$^{*}$ LR values measured at 1064 nm are used for LR at 910 nm.
"

The mass profiles estimated from backscatter coefficients at 1064 nm (in red color) have been added in the revised figures as well.

We have added uncertainties on the backscatter coefficients (Fig.5 a,b). But it will be too messy if we add such information on Fig.5c due to the large uncertainties on the mass concentration. Nevertheless, we have emphasized this in the figure caption.

In addition, we have added a new sub-figure (Fig.5d) to show the difference on the estimated mass concentrations as follows:
"

[Figure]

**Figure 6. (a) Lidar-derived backscatter coefficients (BSC) at 355 (blue), 532 (green), and 1064 nm (red) from Polly[XT], and at 910 nm (black) from CL51. (b) BSCs at 532 nm: measured at 532 nm (meas.), or converted (conv.) from measured BSCs at other wavelengths. (c) Estimated mass concentration profiles for the SPoI (Smoke Plume of Interest) using BSCs in (b), based on parameters in Table 2-method #1. Mass concentrations from MERRA-2 model are also shown in orange colour with corresponding time given on the bottom right of each panel. (d) Relative differences on the mass concentrations (denoted as *m*) estimated from measured/converted BSCs, and of MERRA-2 model, using the one from measured BSC at 532 nm as the reference. 2 h time-averaged lidar profiles are used, with the time slot (UTC) on 5 June 2019 given on top of each panel. The horizontal lines (in a, b) illustrate the uncertainties range. The uncertainties in mass concentrations (in c) are discussed in Sect. 3.2.1.**

"

Concerning the quantitative description on 'mass concentration value', 'good agreement' or 'large discrepancies', we have carefully checked the manuscript and made the modifications in the revised version:
In section 3.2.1 "Method #1: based on BAE & the conversion factor from literature":
"

The peak value of the mass concentrations was found at 6–8h UTC, of ~ 23.5 (27.5) μg m$^{-3}$ estimated from the backscatter coefficients at 532 nm (910 nm). If we take the mass concentration estimated from the BSC at 532 nm as the reference, good agreements are found between the mass concentrations estimated from BSCs at different wavelengths (Fig. 5 d). The mean values of the relative differences were around 8 %, 12 %, and 18 % for the estimations from BSCs at 355, 910 and 1064 nm, respectively. Comparing 532 and 355 nm mass estimates, better agreements were found during daytime (8–20h UTC), with a difference <6 %. Nonetheless, considering

532 and 910 nm estimates, the best agreements were found at 6–8 and 20–24h UTC, with a difference <3 %, whereas the worst agreement of ~ 30 % was found at 14–16h UTC. Larger differences between 910 and 1064 nm estimates were found, with a mean relative difference of ~ 28 %, and a highest value of ~ 64 % at 14–16h UTC.
"

In section 3.2.2 "Method #2: BSC at each wavelength & conversion factors from site":
"

The peak value of the mass concentrations estimated from the BSCs at 532 nm reached ~ 38 µg m$^{-3}$ at 6–8h UTC, higher than the one estimated from method #1 because of the bigger conversion factor. The relative differences on the mass concentrations estimated from the BSCs at different wavelengths were analysed (Fig. 7 b). Similarly, we take the mass concentration estimated from the BSCs at 532 nm (which is the wavelength most often used in earlier studies) as the reference, and found an underestimate when using BSCs at 355 nm, with a mean bias of ~ 15 %, and a peak bias of ~ 25 % at 4–6h UTC; the best agreement was found for night-time measurements (20–24 h UTC) with a bias <5 %. Nevertheless, an overestimate was found for the mass concentration estimated from the BSCs at 910 nm, with a mean bias of ~ 36 %, a peak bias of ~ 68 % at 14–16h UTC, and a minimum bias of ~ 14 % at 10–12h UTC. The overestimate for CL51-derived mass concentrations could be due to an overestimate of LR at 910 nm, since we used LR at 1064 nm in the calculations. In addition, big differences (with a mean value of ~ 42 %) were found between the CL51-derived mass concentrations and the ones estimated from the Polly$^{XT}$-derived BSCs at 1064 nm; highest discrepancy were found of ~ 95 % at 14–16h and ~ 75 % at 16–18h UTC, whereas better agreements were found at 4–6h, 10–12h, and 18–24h, with bias <7 %.
"

In section 3.3 "Comparison with MERRA-2 model - wildfire smoke and dust aerosol mixture":
"

In this section, the MERRA-2 mass concentrations were compared with the mass concentrations estimated from the Polly$^{XT}$ backscatter coefficients at 532 nm (from method #1-Fig. 5 c, and method #2-Fig. 7 a). Note that the main difference on Polly$^{XT}$-estimated mass concentrations from two methods are due to the different conversion factor values (Table 2), thus the mass concentrations estimated from BSCs at 532 nm using method #1 are ~ 40 % lower than method #2. When the Polly$^{XT}$ estimates from method #1 were used as the reference, good consistencies were found in the morning (at 6h, 9h, and 12h UTC), with overestimations (<30 %) of MERRA-2 mass concentrations; whereas large discrepancies were found in the afternoon, with high overestimations of ~ 160 % at 15h UTC and ~ 90 % at 18h UTC. If the Polly$^{XT}$ estimates from method #2 were used as the reference, good consistencies were also found in the morning (at 6h, 9h, and 12h UTC), but with underestimations (<30 %); and a large overestimation of ~ 63 % was found at 15h UTC. At 15h UTC, the MERRA-2 simulated dust mass concentration fraction is more than half of the MERRA-2 simulated total mass concentration. It is good to keep in mind that both observations and simulations have significant uncertainties. The presence of cirrus cloud in the upper atmosphere during the day may also have some impacts on MODIS AOD, which is assimilated by the MERRA-2 model.
"

**Pp 10, l 300. From the text I understand that the fine dust comes from N America. Please comment and justify the presence of dust. I would rather think about Saharan dust as the event described by Osborne et al. (2019). I guess the Hysplit did not show backtrajectories towards N Africa in your case. I saw that MERRA-2 shows a dust component.**

Thank you for the comments. Using Hysplit, we found that the air mass comes from the N America not N Africa. We have checked the AIRS dust score (figure below), it shows that there was dust (inside the red circle in the figure) in N America on the day when the air mass passed by. We have added this information in the revised version for the clarity.
In section 2.1, we added:
"

The "Dust score" data provided by AIRS (Atmospheric InfraRed Sounder) were used to determine the occurrences of dust events (https://airs.jpl.nasa.gov, last access: 1 July 2021).
"

In section 3, we added:
"

The backward trajectory analysis was performed using the HYSPLIT model. The analysis shows that particles in the SPoI had travelled about seven days from the forest fire sources (MODIS, 2019) in western Canada to North Europe (Fig. 4). The AIRS dust score map (https://airs.jpl.nasa.gov/map/, last access: 1 July 2021) also showed some dust presence in North America on 30 May.

"

[Figure]

[Figure]

May 30, 2019

**AIRS dust**

**Pp 12, 357-359: taking into account the uncertainties in the retrieval of the mass concentration, the improvement by 4% using dust contribution seems not very relevant.**

We agree, so we have removed the Fig.6c (of old version) as it shows only few information.
We performed this separation study in order to check how the inclusion of dust (as indicated by MERRA-2) would affect the mass concentration estimations. We found out that the inclusion of a dust mixture results in slightly

higher estimated mass concentration values, with a difference negligible considering the uncertainties. Thus, we conclude that lidar and ceilometer observations for mass retrievals can be used even without exact knowledge on the composition of the smoke plume in the troposphere.

We have clarified this in the revised version:

In section 3.3 "Comparison with MERRA-2 model - wildfire smoke and dust aerosol mixture":

"

The mass concentrations from MERRA-2 model data are used for the comparison with the lidar retrievals. An interesting feature in the MERRA-2 simulation results is the presence of dust in the SPoI. The contribution of dust to the total AOD is very low (much lower than the carbon optical depth), indicating that the dust particles are in the fine mode. However, the dust contribution to the total mass concentration is non-negligible. Low values of lidar-derived depolarization ratio suggest no significant presence of non-spherical particles, but in principle, a small amount of dust could be mixed with the smoke. It is possible that there are biomass burning aerosols and fine dust aerosols in the SPoI, as only fine dust particles should be able to remain long enough in the atmosphere to be transported from North America to Kuopio. Furthermore, the air masses in SPoI passed by the area in North America where dust was present (shown by the AIRS data).

…

In order to check how the inclusion of dust (as indicated by MERRA-2) would affect the mass concentration estimations, we assume that there were wildfire smoke and fine dust aerosol mixture in the SPoI. The POLIPHON (Polarization lidar photometer networking) method (Mamouri and Ansmann, 2014, 2017) was applied to separate fine dust (particles with radius < 500 nm) and biomass burning aerosols for the SPoI.

…

For the example given in Fig. 8, the fine dust contributes ~ 13 % to the extinction in the SPoI, whereas its mass concentration contributes ~ 32 % (method #1) or ~ 23 % (method #2) to the total mass concentration. However, the derived total mass concentration considering a fine dust and smoke mixture is only ~ 18 % (method #1) or ~ 4 % (method #2) higher than one assuming smoke particles only. The inclusion of a dust mixture results in slightly higher estimated mass concentration values, with a difference negligible considering the uncertainties. We have also performed POLIPHON considering coarse mode dust mixture; higher (~20–30 %) total mass concentrations were retrieved but still within the uncertainty range. The aged smoke aerosols may also introduce enhanced depolarization ratios. If we use a bigger value (e.g., 0.05) instead of 0.03 as the smoke depolarization ratio in POLIPHON, the dust impacts on the mass concentration estimations are even smaller. Hence, the mass estimations of the SPoI considering only smoke are good enough even if the plume contains small amount of dust.

Similar conclusion can also be applied to ceilometer observations. It is not possible to perform the aerosol separation using ceilometer data alone, as no depolarization information is available at this wavelength. For this instrument, only one aerosol type should always be assumed in the layer of interest, which then imparts an additional bias when estimating the mass concentration. However, we have shown in this section that ceilometer observations for mass retrievals can be used even without exact knowledge on the composition of the smoke plume in the troposphere.

"

**Change units from um to ug.**

The correction has been done.

**Fig 5. Please add uncertainties to profiles.**

We have removed this figure and also the text about in situ measurements in the revised version.

**Fig. 7 please add uncertainties to profiles.**

We have added the uncertainties.

[Figure]

**Figure 8. Lidar products obtained from Polly$^{XT}$ measurements on 5 June 2019, 8–10h UTC (2h signal average). (a) Measured 532 nm total particle backscatter coefficient (green) and particle linear depolarization ratio (brown). (b) Particle backscatter coefficients (BSCs) for fine dust (orange) and smoke (blue) particles, obtained with the POLIPHON method. (c) Respective fine dust and smoke extinction coefficients (EXTs) obtained by multiplying the BSCs (in b) with the lidar ratios. (d,e) The fine dust (orange), smoke (blue) particle mass concentrations derived from the EXT profiles (in c), by using parameters in Table 4 for method #1 (d) and #2 (e). The total mass concentrations of fine dust and smoke mixture (purple), or of only smoke particles (green) are shown. (f) Mass concentrations of organic carbon (OC, dark green), black carbon (BC, light green), dust (orange), sea salt and sulphate (SS+SU, red) from MERRA-2 model, at 9h UTC on 5 June. The total mass concentration profile is also given by orange squares. The horizontal lines illustrate the uncertainties range.**

**Response to Referee #2**

Thank you for carefully reading the manuscript and providing useful suggestions to improve the paper. The replies to the referee comments are given below. The referee comments are highlighted in blue with our responses in black. The sentences in the manuscript are between the quotation marks, with the modifications in the revised manuscript in red.

**Authors provide comparison of the particle mass concentration obtained from Raman lidar and from ceilometer measurements, which is important topic. The manuscript is well written and can be published after minor revision.**

We are grateful to the referee for positive assessment of our work.

**Ln 159. "calculated from the relative humidity and the temperature profiles from GDAS1 data"**

**GDAS profiles may differ significantly from real profiles of the water vapor. This should be kept in mind when using these for correction. Did you compare GDAS with profiles obtained from Raman measurements?**

Thank you for the comment. We agree with the referee that GDAS profiles may differ from real ones. Unfortunately, radiosonde measurements were not collocated with lidar observations. The nearest sounding station locates at Jokioinen, which is ~300 km away from our measurement site. The Raman lidar relative humidity measurements are only available at nighttime. Nevertheless, we compared GDAS profiles with Raman lidar profiles, good agreements were found.
We have added in section 2.1 in the revised version for the clarity:
"

Temperature and pressure profiles from the GDAS (Global Data Assimilation System, https://www.ready.noaa.gov/gdas1.php, last access: 19 March 2021) database were used for the correction of Rayleigh extinction and backscattering effects for lidar data analysis. Aiming at the observations of water vapor profiles, the most used and well-established measurement method is radiosonde sounding. However, the closest available radiosonde data are from Jokioinen (Finland), located ~ 300 km away from the measurement site. Filioglou et al. (2017) reported the inadequate vertical representation of water vapor due to the non-stable atmospheric conditions between two sites, when using a radiosonde 100 km away. Thus, the relative humidity profiles from GDAS data were used for the water vapor number densities estimations.

"

**Ln.166. What can we conclude from using both forward and backward Klett methods? Forfward method is very sensitive to the choice of lidar ratio. The lidar ratio of smoke can vary in significant range, so use of just one value obtained from rotational Raman lidar is risky. Did you compare it with lidar ratio provided by AERONET?**

Thank you for the comments. We agree that the extinction retrieval is sensitive to the assumed lidar ratio. In this study we have chosen the reference height as close as to the layer of interest, so that the error propagation could be minimized. This was clarified in the revised version. We have also added description about forward and backward method in the revise version.
In section 2.3:
"

The retrieval methods for deriving the backscatter coefficient from ceilometers are quite mature (Wiegner et al., 2014; Wiegner and Geiß, 2012). Under favourable conditions, a relative error of the backscatter coefficient on the order of 10 % seems feasible with a careful calibration by applying the forward integration. On the contrary, significant temporal averaging of ceilometer data is required for performing a Rayleigh calibration, as the detection of molecular signals is intrinsically very difficult. Binietoglou et al. (2011) propose a two-step approach, resulting promising agreement comparing to their lidar PEARL (Potenza EARLINET Raman lidar). The uncertainty of the backscatter coefficient could be in the range of 20–30 % using the backward integration. The advantage of the forward algorithm is that calibration is required only occasionally, and it is not affected by the low SNR in the upper troposphere. However, the accuracy in deriving extinction coefficients is limited due to

the unknown LR at 910 or 1064 nm and its uncertainties. In particular the presence of multi-layered aerosol distributions (with different aerosol types) may introduce more uncertainties. In addition, the uncertainty due to the neglecting the water vapor increased with the distance from the chosen reference height. In this study, we applied the Klett method (Wiegner et al., 2014) by defining the reference height as close as to the layer of interest, so that the error propagation (due to uncertainties of LR and water vapor transmission) would be minimized for that layer.
"

The lidar ratio provided by AERONET is for the total atmospheric column (as we mentioned in section 2.1), during the period, there were multi aerosol layers. Thus, it is not fully comparable using AERONET LR and PollyXT LR of smoke layer. Besides, AERONET level 2.0 LRs were not available on 5 June (the day of the SPoI - smoke layer).

**Fig.1c. Signal above 3 km is very noisy so results probably depends on choice of the reference height.**

We agree, so we chose the reference height as close as to the layer of interest to minimize the error. We have added the uncertainties due to wrong assumptions of $\lambda_0 \pm 2$ nm and due to the analytical solution in the Fig. 1.
"

[Figure]

Figure 1. Example of water vapor corrections on 2 h averaged ceilometer data on 5 June 2019 (20:00–22:00 UTC). (a) Relative humidity (RH, teal) and water vapor number density ($n_w$, brown) from GDAS1 data at 21:00 UTC. (b) Range-corrected signal at 910 nm, without (RCS*, red) or with (RCS, black) water vapor correction, and the hypothetical Rayleigh-signal at 910 nm (dashed blue). (c) Retrieved particle backscatter coefficients: β* without (red) and β with (black) water vapor correction, using forward (FW) integration Klett solution. (d) Same as (c) but application of the backward (BW) integration. (e) Ratio of the retrieved β* and β, when using forward integration (magenta), or backward integration (green). The horizontal lines illustrate the uncertainties range due to wrong assumptions of the central wavelength $\lambda_0 \pm 2$ nm. The uncertainties in backscatter coefficients of the analytical solution were shown by dashed lines.

"

**Fig.5d. I am a little confused. What means "contribution of BrC"? Please specify.**

It is the brown carbon (BrC) contribution to the absorption coefficient (at 370 nm).
But we have removed this figure and also the text about in situ measurements in the revised version.

**Fig.6. The demonstration of the temporal evolution of the profiles is useful. Still would be good to quantify the difference between lidar and ceilometer. May be provide standard deviation?**

Thank you for the suggestion, we have added a new sub-figure (Fig.5d) to show the difference on the estimated mass concentrations as follows.
"

[Figure]

**Figure 6. (a) Lidar-derived backscatter coefficients (BSC) at 355 (blue), 532 (green), and 1064 nm (red) from Polly[XT], and at 910 nm (black) from CL51. (b) BSCs at 532 nm: measured at 532 nm (meas.), or converted (conv.) from measured BSCs at other wavelengths in (a). (c) Estimated mass concentration profiles for the SPoI (Smoke Plume of Interest) from BSCs in (b), based on parameters in Table 2-method #1. Mass concentrations from MERRA-2 model are also shown in orange color with corresponding time given on the bottom right of each panel. (d) Relative differences on the mass concentrations (denoted as *m*) estimated from measured/converted BSCs, and from MERRA-2 model, using the one estimated from measured BSC at 532 nm as the reference. 2 h time-averaged lidar profiles are used, with the time slot (UTC) on 5 June 2019 given on top of each panel. The horizontal lines (in a, b) illustrate the uncertainties range. The uncertainties in mass concentrations (in c) are discussed in Sect. 3.2.**

"

**I agree with the first reviewer, that numbers should be used instead expressions like "good agreement".**

Thank you for the suggestion, we have carefully checked the manuscript and made modifications for the quantitative description on 'mass concentration value', 'good agreement' or 'large discrepancies', in the revised version:

In section 3.2.1 "Method #1: based on BAE & the conversion factor from literature":

"

The peak value of the mass concentrations was found at 6–8h UTC, of ~ 23.5 (27.5) μg m$^{-3}$ estimated from the backscatter coefficients at 532 nm (910 nm). If we take the mass concentration estimated from the BSC at 532

nm as the reference, good agreements are found between the mass concentrations estimated from BSCs at different wavelengths (Fig. 5 d). The mean values of the relative differences were around 8 %, 12 %, and 18 % for the estimations from BSCs at 355, 910 and 1064 nm, respectively. Comparing 532 and 355 nm mass estimates, better agreements were found during daytime (8–20h UTC), with a difference <6 %. Nonetheless, considering 532 and 910 nm estimates, the best agreements were found at 6–8 and 20–24h UTC, with a difference <3 %, whereas the worst agreement of ~ 30 % was found at 14–16h UTC. Larger differences between 910 and 1064 nm estimates were found, with a mean relative difference of ~ 28 %, and a highest value of ~ 64 % at 14–16h UTC.

"

In section 3.2.2 "Method #2: BSC at each wavelength & conversion factors from site":

"

The peak value of the mass concentrations estimated from the BSCs at 532 nm reached ~ 38 $\mu g\ m^{-3}$ at 6–8h UTC, higher than the one estimated from method #1 because of the bigger conversion factor. The relative differences on the mass concentrations estimated from the BSCs at different wavelengths were analysed (Fig. 7 b). Similarly, we take the mass concentration estimated from the BSCs at 532 nm (which is the wavelength most often used in earlier studies) as the reference, and found an underestimate when using BSCs at 355 nm, with a mean bias of ~ 15 %, and a peak bias of ~ 25 % at 4–6h UTC; the best agreement was found for night-time measurements (20–24 h UTC) with a bias <5 %. Nevertheless, an overestimate was found for the mass concentration estimated from the BSCs at 910 nm, with a mean bias of ~ 36 %, a peak bias of ~ 68 % at 14–16h UTC, and a minimum bias of ~ 14 % at 10–12h UTC. The overestimate for CL51-derived mass concentrations could be due to an overestimate of LR at 910 nm, since we used LR at 1064 nm in the calculations. In addition, big differences (with a mean value of ~ 42 %) were found between the CL51-derived mass concentrations and the ones estimated from the Polly$^{XT}$-derived BSCs at 1064 nm; highest discrepancy were found of ~ 95 % at 14–16h and ~ 75 % at 16–18h UTC, whereas better agreements were found at 4–6h, 10–12h, and 18–24h, with bias <7 %.

"

In section 3.3 "Comparison with MERRA-2 model - wildfire smoke and dust aerosol mixture":

"

In this section, the MERRA-2 mass concentrations were compared with the mass concentrations estimated from the Polly$^{XT}$ backscatter coefficients at 532 nm (from method #1-Fig. 5 c, and method #2-Fig. 7 a). Note that the main difference on Polly$^{XT}$-estimated mass concentrations from two methods are due to the different conversion factor values (Table 2), thus the mass concentrations estimated from BSCs at 532 nm using method #1 are ~ 40 % lower than method #2. When the Polly$^{XT}$ estimates from method #1 were used as the reference, good consistencies were found in the morning (at 6h, 9h, and 12h UTC), with overestimations (<30 %) of MERRA-2 mass concentrations; whereas large discrepancies were found in the afternoon, with high overestimations of ~ 160 % at 15h UTC and ~ 90 % at 18h UTC. If the Polly$^{XT}$ estimates from method #2 were used as the reference, good consistencies were also found in the morning (at 6h, 9h, and 12h UTC), but with underestimations (<30 %); and a large overestimation of ~ 63 % was found at 15h UTC. At 15h UTC, the MERRA-2 simulated dust mass concentration fraction is more than half of the MERRA-2 simulated total mass concentration. It is good to keep in mind that both observations and simulations have significant uncertainties. The presence of cirrus cloud in the upper atmosphere during the day may also have some impacts on MODIS AOD, which is assimilated by the MERRA-2 model.

"

**Response to Referee #3**

Thank you for carefully reading the manuscript and providing useful suggestions to improve the paper. The replies to the referee comments are given below. The referee comments are highlighted in blue with our responses in black. Some comments concerning similar issues are grouped together. The sentences in the manuscript are between the quotation marks, with the modifications in the revised manuscript in red.

**General**

**The paper contains smoke observations over Finland. But this aspect alone is, to my opinion, not sufficient to justify publication. Meanwhile there are so many smoke observations with lidar in the literature (see review of Adam et al., 2020) and even over the North Pole (Ohneiser et al., 2021). Therefore, the goals of the paper need to better emphasized: lidar-ceilometer observations and comparison with model results is probably one goal. Another goal is the careful analysis (some kind of a feasibility study) to what extent Vaisala ceilometers (and these huge ceilometer networks) can contribute to tropospheric smoke monitoring (even in terms of mass concentration profiling). The paper is worthwhile to be published, however only after significant improvement. Furthermore, the paper contains many speculative and questionable aspects. Their own AERONET approach to derive smoke conversion factors is unacceptable. So, there are many parts that need to be significantly improved.**

**Major revisions are required.**

**Details:**

**Abstract:**

**It should be clearly stated in the beginning: What is the main goal of the paper, what is new in this paper (in view of the numerous smoke observations with lidar in the literature, see review of Adam et al., 2020). First lidar smoke observations over Finland …. is not a convincing argument (or goal). Recently, TROPOS people even measured smoke over the North Pole (Engelmann et al., 2020, Ohneiser et al., 2021) … with lidar aboard an ice breaker.**

**To my opinion, to combine lidar and ceilometer observations (and even to include modelling) is an attractive approach. And especially, if the main goal is: … to demonstrate the usefulness of a Vaisala ceilometer to monitor smoke in the troposphere!**

**However, feel free to define your specific goals! This is just a suggestion. In this context, you can then easily present all your nice smoke results on changing depolarization ratios, on this unique smoke feature with larger lidar ratios at 532 than at 355 nm, and the comparison with model results for smoke.**

Thank you for the suggestion.
We have changed the title of the manuscript to better communicate the main goal of our analysis:
"
Mass concentration estimates of long-range-transported Canadian biomass burning aerosols from a multi-wavelength Raman polarization lidar and a ceilometer in Finland
"
We made modifications in abstract in the revised version and emphasized the goal of the manuscript:
"
A quantitative comparison study for Raman lidar and ceilometer observations, and for model simulations of mass concentration estimates of smoke particles is presented. Layers of biomass burning aerosol particles were observed in the lower troposphere, at 2 to 5 km height on 4 to 6 June 2019, over Kuopio, Finland. These long-range-transported smoke particles originated from a Canadian wildfire event. The most pronounced smoke plume detected on 5 June was intensively investigated. Optical properties were retrieved from the multi-wavelength Raman polarization lidar Polly$^{XT}$. Particle linear depolarization ratios (PDR) of this plume were measured to be $0.08 \pm 0.02$ at 355 nm and $0.05 \pm 0.01$ at 532 nm, suggesting the presence of partly coated soot particles or particles that have mixed with a small amount of dust or other non-spherical aerosol type. The layer-mean PDR at 355 nm (532 nm) decreased during the day, from ~ 0.11 (0.06) in the morning to ~ 0.05 (0.04) in

the evening; this decrease with time could be linked to the particle aging and related changed in the smoke particle shape properties. Lidar ratios were derived as 47 ± 5 sr at 355 nm and 71 ± 5 sr at 532 nm. A complete ceilometer data processing for a Vaisala CL51 is presented, including the water vapor correction for high latitude for the first time, from sensor provided attenuated backscatter coefficient to particle mass concentration. Aerosol backscatter coefficients (BSCs) were measured at four wavelengths (355, 532, 1064 nm from Polly$^{XT}$, and 910 nm from CL51). Two methods, based on a combined lidar and sun-photometer approach, are applied for mass concentration estimations from both Polly$^{XT}$ and the ceilometer CL51 observations. In the first method #1 we used converted BSCs at 532 nm (from measured BSCs) by corresponding measured backscatter-related Ångström exponent, whereas in the second method #2 we used measured BSCs at each wavelength independently. A difference of ~ 12 % or ~ 36 % was found between Polly$^{XT}$ and CL51 estimated mass concentrations using method #1 or #2, showing the potential of mass concentration estimates from ceilometer. Ceilometer estimations have uncertainty of ~ 50 % in the mass retrieval, but the potential of the data lays in the great spatial coverage of these instruments. The mass retrievals were compared with the Modern-Era Retrospective analysis for Research and Applications, version 2 (MERRA-2) meteorological and aerosol reanalysis. The inclusion of dust (as indicated by MERRA-2 data) in the retrieved mass concentration is negligible considering the uncertainties, which also shows that ceilometer observations for mass retrievals can be used even without exact knowledge on the composition of the smoke dominant aerosol plume in the troposphere.

"

**Introduction:**

**P 2-3: The Introduction should be improved. You mention the Mueller1999 paper, then I would add the Mueller2005 paper as well because that paper is directly related to smoke observations (and lidar inversion application). Furthermore, you need to mention this Adam 2020 review paper!**

**In the next step, you may want to continue with network activities (before you introduce the ceilometer network aspect), and maybe, also CALIPSO observations. Again, with clear focus on smoke. There are these Baars2019 and Khaykin2018 papers as examples for network and space lidar activities. This would show the added value towards regional to global scale smoke characterization when using networks. This motivates, to my opinion, then the next step: …. to analyse to what extent the existing and exciting (European) ceilometer infrastructure could do in case of smoke monitoring… and so on…. All this would corroborate the importance of the paper. Are there some smoke observations with ceilometers in the literature (I am not sure)? If yes, should be cited. If not, that would be new point to be mentioned! One may also indicate similar approaches such as the ceilometer observations of volcanic aerosols (Eyjafjalla volcanic aerosol, Emeis and Flentje papers in 2010/2011?) to indicate the usefulness of modern ceilometers to detect aerosols (and not only clouds).**

**Feel free to define your own specific goals of the paper. It is not very clear to me at the moment what the goals are.**

Thank you for the useful comment and detailed suggestions, we have enriched our introductions, and emphasized the goal of the manuscript:
"

…

[revised manuscript text omitted]

"

**Now some more detailed remarks:**

**P6, line 176: I do not believe that you can get the backscatter coefficient at 910 or 1064 nm with an uncertainty of less than 10%. The uncertainty in the reference value is too large. And a proper Rayleigh fit at these long wavelengths almost impossible. The uncertainty is certainly always in the range of 20-30% at 910 or 1064 nm for the backscatter coefficient.**

Thank you for pointing it out, we have made modifications in the revised version.
"

The retrieval methods for deriving the backscatter coefficient from ceilometers are quite mature (Wiegner et al., 2014; Wiegner and Geiß, 2012). Under favourable conditions, a relative error of the backscatter coefficient on the order of 10 % seems feasible with a careful calibration by applying the forward integration. On the contrary, significant temporal averaging of ceilometer data is required for performing a Rayleigh calibration, as the detection of molecular signals is intrinsically very difficult. Binietoglou et al. (2011) propose a two-step approach, resulting promising agreement comparing to their lidar PEARL (Potenza EARLINET Raman lidar). The uncertainty of the backscatter coefficient could be in the range of 20–30 % using the backward integration. The advantage of the forward algorithm is that calibration is required only occasionally, and it is not affected by the low SNR in the upper troposphere. However, the accuracy in deriving extinction coefficients is limited due to the unknown LR at 910 or 1064 nm and its uncertainties. In particular the presence of multi-layered aerosol distributions (with different aerosol types) may introduce more uncertainties. In addition, the uncertainty due to the neglecting the water vapor increased with the distance from the chosen reference height. In this study, we applied the Klett method (Wiegner et al., 2014) by defining the reference height as close as to the layer of interest, so that the error propagation (due to uncertainties of LR and water vapor transmission) would be minimized for that layer.

"

**And the conversion (backscatter to extinction) will introduce another 20-40% uncertainty in the case of smoke layers. The lidar ratio for smoke was found to be 50, 60, 70, 80, even 110 sr in smoke observation (see Adam et al, including the ACPD version and supplementary tables). So, using, e.g., 75 sr as smoke lidar ratio at 532 nm, and the range is from 50 to 100 sr, than the error is 33%. The uncertainties are probably similar for 910 nm.**

**P10, lines 288-289, please re-calculate the uncertainties by assuming 20% (BSC), 30% (LR), 20% (conversion factor from literature) and 20% (particle density), probably the uncertainty is 40-50%.**

Thank you for the comment. We have re-calculated the uncertainties and made modifications in the manuscript; we also added a new Table 3 for the clarity as follows in the revised version.
In section 2.2 "PollyXT lidar":
"

The relative uncertainties are in the range of 5–10 % for backscatter coefficients and depolarization ratios at 355 and 532 nm (Ansmann et al., 1992b; Baars et al., 2012). The backscatter coefficients retrieval at 1064 nm may be possible with a relative uncertainty of 15 % using only elastic signal by assuming a proper lidar ratio. The lidar ratios at 355 and 532 nm are measured with a typical relative uncertainty of ~ 20 % when the inelastic

measurements are good enough. Higher uncertainties in lidar ratio at 1064 nm (~ 30 %) should be considered (Haarig et al., 2018).

"

**Table 3. Relative uncertainties in the input parameters and in the retrieved products (in bold). The uncertainty origins are given for input parameters and denoted as: R-Raman measurement available, E-only elastic measurement for the retrieval, L-literature, A-assumption. The uncertainty in the smoke mass density ($\rho$) was assumed as 20 % as in Ansmann et al. (2021). Different retrieval information (R or E) is available at each wavelength with a different system (Polly$^{XT}$ or CL51), thus different uncertainties in the backscatter coefficients ($\beta$) and lidar ratio ($LR$) are considered. The uncertainty in the smoke volume-to-extinction conversion factor ($c_v$) was assumed as 10 % for both methods, as given in Ansmann et al. (2021). The relative uncertainties in the mass concentration ($m$), backscatter-related Ångström exponent ($BAE$), and converted backscatter coefficient ($\beta_{conv}^{532}$) are obtained by the error propagation applied to Eqs. 1-5.**

| | | | Polly$^{XT}$ | | CL51 |
|---|---|---|---|---|---|
| | $\lambda$ (nm) | 532 | 355 | 1064 | 910 |
| | Uncertainty | | | | |
| Common | $\Delta\rho/\rho$ | | 0.20 (L) | | |
| | $\Delta\beta/\beta$ | 0.10 (R) | 0.10 (R) | 0.15 (E) | 0.20 (E) |
| Method #1 | $\Delta c_v/c_v$ | 0.10 (L) | - | - | - |
| if $\beta$ (532) | $\Delta LR/LR$ | 0.20 (R) | - | - | - |
| available | $\Delta\boldsymbol{BAE}/\boldsymbol{BAE}$ * | - | **0.14** | **0.12** | **0.24** |
| | $\Delta\boldsymbol{\beta_{conv}^{532}}/\boldsymbol{\beta_{conv}^{532}}$ ** | - | **0.18** | **0.24** | **0.31** |
| | $\Delta\boldsymbol{m}/\boldsymbol{m}$ | **0.32** | **0.36** | **0.40** | **0.45** |
| if $\beta$ (532) | $\Delta LR/LR$ | 0.30 (A) | - | - | - |
| not available | $\Delta\boldsymbol{BAE}/\boldsymbol{BAE}$ * | - | 0.30 (A) | 0.30 (A) | 0.30 (A) |
| | $\Delta\boldsymbol{\beta_{conv}^{532}}/\boldsymbol{\beta_{conv}^{532}}$ ** | - | **0.33** | **0.51** | **0.36** |
| | $\Delta\boldsymbol{m}/\boldsymbol{m}$ | - | **0.67** | **0.52** | **0.54** |
| Method #2 | $\Delta c_v/c_v$ | 0.10 (A) | 0.10 (A) | 0.10 (A) | 0.10 (A) |
| | $\Delta LR/LR$ | 0.20 (R) | 0.20 (R) | 0.30 (L) | 0.40 (A) |
| | $\Delta\boldsymbol{m}/\boldsymbol{m}$ | **0.32** | **0.32** | **0.42** | **0.52** |

* Wavelength pair of λ and 532, ** Converted backscatter coefficient at 532 nm from λ.

We have added discussions on uncertainty studies in the revised version:
In section 3.2.1 for the method #1:
"

In Table 3, the uncertainties in the input parameters and the estimated mass concentrations are listed. We assume an uncertainty of 20 % in the smoke mass density (Ansmann et al., 2021). The uncertainties in backscatter coefficients at different wavelengths and lidar ratio at 532 nm follow from the discussions in Sect. 2. The conversion factor and lidar ratio at 532 nm are required as input, with assumed uncertainties of 10 % (given in Ansmann et al., 2021) and 20 % (c.f., Sect. 2.2), respectively. The uncertainties in BAE between different wavelength pairs, and in $\beta_{conv}^{532}$ were obtained by error propagations to Eqs. (3,4). Note that the standard deviations of BAE from our measurements (Table 2) show lower values than their uncertainties. Finally, after applying the law of error propagation to Eq. (5), we expect an overall uncertainty in the mass concentration estimates of 32–45 %. The highest uncertainty of 45 % was found when using the ceilometer method, mainly due to the higher uncertainty of 20 % in the backscatter coefficient retrieval.

However, the lidar measurements at 532 nm are not always collocated, especially for numerous ceilometer stations. For those cases, the lidar ratio at 532 nm and the BAEs (or colour ratios) should be assumed, thus with higher uncertainties. We can assume uncertainties of 30 % in lidar ratio at 532 nm and 30 % in BAEs for all

wavelength pairs, thus, the uncertainty for the estimated mass concentrations will be over 50 % (Table 2). For the smoke particles, extended overviews of observed wavelength dependencies of backscatter coefficients can be found in Burton et al. (2012) and Adam et al. (2020).

,,

In section 3.2.2 for the method #2:

"

The uncertainties in the input parameters and the estimated mass concentrations of this method #2 are listed in Table 3. The uncertainties in the conversion factors from the standard deviation in Table 2 are very small due to the limited sample number, thus 0.10 was used as proposed in Ansmann et al. (2021). The uncertainties in backscatter coefficients and lidar ratios at each wavelength follow from the discussion in Sect. 2. Uncertainties in the lidar ratio at ceilometer wavelengths are much larger, particularly, as we applied the lidar ratio value measured at 1064 nm to the ceilometer wavelength of 910 nm. Thus, we assume an uncertainty of 40 % in the ceilometer lidar ratio. The overall uncertainties in the mass concentration estimates are of about 30–50 %, with the highest uncertainty of 52 % when using ceilometer measurements.

As can be seen in Table 3, when applying the method #2, the uncertainty in mass concentration estimations is slightly lower using measured BSCs at 355 nm, whereas higher uncertainties were found when using measured BSCs at 1064 and 910 nm. The main reason lies in the high uncertainties in lidar ratios at 1064 and 910 nm. Hence, when the lidar ratio can be measured or properly estimated, and the conversion factor can be estimated under the pure aerosol type condition, method #2 is recommended. Otherwise, method #1 can be applied by using properly estimated BAEs or colour ratios.

,,

**P7, lines 201 – 214: I speculate that there was an air mass transport from central Europe to Finland at heights below 3 km height (not presented), when I see the backward trajectory figure for the arrival height of 4 km. And this aged European haze widely determined the observed AOD over the field sites. You mention 500 nm AODs of 0.24-0.42 (as written on page 6). And for the smoke layer the 532 nm AOD was found to be 0.02 to 0.13 (page 7, Sect 3.1.). So the smoke impact was at least not dominating. This means that the AERONET observations cannot be used to derive smoke conversion factors. This point will be further discussed below.**

**P9, line 256: It makes no sense to me at all to use the actual AERONET data to derive smoke conversion factors. As mentioned, the AOD was obviously dominated by European pollution, so that the conversion factors reflect European fine mode haze properties. All the efforts to get proper conversion factors from AERONET (dust, smoke, marine, etc.) were done in regions with pure dust or marine or smoke conditions, etc. One should therefore use the conversion factors presented in this Ansmann 2020 paper, or you try to use the Polly multiwavelength information (inversion) to obtain the smoke volume concentration in the smoke layer together with the backscatter and extinction coefficients in these layers, and in this way the required smoke conversion parameters. Your conversion factor of 0.21 perfectly describes the conversion factor for urban haze. The smoke conversion factors are in the range from 0.12-0.15, and thus considerably lower.**

Thank you for the comments.

The smoke layer AOD only present the partial smoke AOD for the smoke on 5 June, as the values are for the selected layer (SPoI), whereas there were several layers on those days. Unfortunately, the AERONET level 2 inversion products were not available on 5 June (when there was the SPoI – selected layer), we therefore used the AERONET data on 6 June for the factor estimate in the manuscript.

We have added one new section 3.2.1 using the "Method #1", where we applied the Ångström exponent to convert the measure backscatter coefficients to 532 nm, and used the conversion factor of 0.13 factor from literature (Ansmann et al. 2021).

We performed more analysis for the second day 6 June (when we used the AERONET factor), and we found that there could be pollution contamination on 6 June, indicated from both back trajectory and CALIPSO observations. We have added this possible inappropriate issue in the revised version.

The method used in the original manuscript is referred as method #2. The possible mixing with pollution in method #2 is now acknowledged in the revised version, and we agree with the reviewer that this is an important source of uncertainty in our analysis. However, although our observations have uncertainties, this method #2 would be the optimal way to estimate aerosol mass when good quality observations are available. Thus, we would like to keep it in the manuscript so that the readers can find information on all the possible ways to do the retrieval in the same paper.

In the revised version two methods are presented, and results from both methods are discussed. We made significant improvement concerning the mass concentration estimations (section 3.2 and 3.3) in the revised version. We didn't copy all the modifications in this reply file, please check these sections (3.2 and 3.3) in the revised version of manuscript.

3.2 Mass concentration estimation
 3.2.1 Method #1: based on BAE & the conversion factor from literature
 3.2.2 Method #2: BSC at each wavelength & conversion factors from site
3.3 Comparison with MERRA-2 model - wildfire smoke and dust aerosol mixture

We introduce two methods in section 3.2 "Mass concentration estimation":
"
This approach was applied to both Polly$^{XT}$ and CL51 data to estimate the mass concentration profiles for biomass burning aerosols in the SPoI. Adapting from the methods describe by Ansmann et al. (2021), we applied two methods in this study:
 Method #1: Mass concentrations were estimated from the measured backscatter coefficients which were converted to 532 nm, using the corresponding measured backscatter-related Ångström exponent. The volume-to-extinction conversion factors at 532 nm from literature was applied (currently the only available wavelength for the smoke factor in the literature).
 Method #2: Mass concentrations were estimated from measured backscatter coefficients at each wavelength of 355, 532, 1064 and 910 nm. The volume-to-extinction conversion factors were evaluated at corresponding wavelengths using AERONET data.
In this study, we assume that both methods can be applied appropriately, and the limitations and sources of uncertainties of method #2 will be discussed in Sect. 3.2.2. The recommendation on the chosen method will be discussed later.
"

We have modified the tables concerning two methods:

**Table 2. Parameters required for the mass concentration retrieval using two methods. The smoke mass density and lidar ratio at 532 nm are common parameters required for both methods #1 and #2.**

| | Parameter | Wavelength | Value | References |
|---|---|---|---|---|
| Common | Smoke mass density (g cm$^{-3}$) | - | 1.3 | Ansmann et al. (2021) |
| | Lidar ratio (sr) | 532 | $71 \pm 5$ | This study |
| Method #1 | Smoke volume-to-extinction conversion factor $c_v$ ($10^{-6}$ m) | 532 | $0.13 \pm 0.01$ | Ansmann et al. (2021) |
| | Backscatter-related Ångström exponent | 355/532 | $2.5 \pm 0.2$ | This study |
| | | 1064/532 | $2.2 \pm 0.3$ | |
| | | 910/532 | $1.8 \pm 0.2$ | |
| Method #2 | Lidar ratio (sr) | 355 | $47 \pm 5$ | This study |
| | | 1064 | $82 \pm 22$ | Haarig et al. (2018) |
| | | 910 | $82 \pm 22^*$ | Haarig et al. (2018) |
| | Fine-mode volume-to-extinction conversion factor $c_v$ ($10^{-6}$ m) | 355 | $0.100 \pm 0.002$ | This study |
| | | 532 | $0.211 \pm 0.003$ | (Possible pollution contamination) |
| | | 910 | $0.620 \pm 0.002$ | |
| | | 1064 | $0.902 \pm 0.004$ | |

$^*$ LR values measured at 1064 nm are used for LR at 910 nm.

**Table 4. Parameters required for the mass concentration retrieval, considering fine dust and smoke mixture.**

| | | Smoke | | Fine dust | |
|---|---|---|---|---|---|
| Mass density (g cm$^{-3}$) | | 1.3 | (Ansmann et al., 2021) | 2.6 | (Ansmann et al., 2012) |
| Depolarization ratio at 532 nm | | 0.03 | (Haarig et al., 2018) | 0.16 | (Sakai et al., 2010) |
| Lidar ratio at 532 nm (sr) | | 71 | this study | 40 | (Ansmann et al., 2019) |
| Volume-to-extinction conversion | method #1 | 0.13 | (Ansmann et al., 2021) | 0.22 | (Ansmann et al., 2019) |
| factor $c_v$(532 nm) (10$^{-6}$ m) | method #2 | 0.21 | this study | 0.22 | (Ansmann et al., 2019) |

The goal of this method #2 is describe at the beginning of section 3.2.2. It requires measurement at only one wavelength, which would be useful for other ceilometer station where there is no lidar measurements at 532 nm.
"

The method #1 is recommended when the measurements at 532 nm are additionally available, or the BAE (or backscatter colour ratio) can be reasonably assumed. Nevertheless, here we suggest a second method, in which mass concentrations were estimated from measured backscatter coefficients at several wavelengths independently, and the measurement at one single wavelength (e.g., for elastic lidars and ceilometers) is required as input for each estimate. This method #2 is recommended in the regions with the pure aerosol type (dust, smoke, marine, etc) condition, where the conversion factor can be evaluated with high accuracy. The mass estimations of the SPoI from measured backscatter coefficients at each wavelength are compared in this section.
"

We have added discussions on the estimated conversion factor, and point out the uncertainties in the factor used in this method #2. We also show the possible pollution contamination for the factor.
"

The estimated conversion factor value at 532 nm of 0.211 ± 0.003 ×10$^{-6}$ m is higher than what we used in the previous section, with the difference ($\Delta c_v$=0.08 ×10$^{-6}$ m) larger than the uncertainty. This value is higher than the values for both fresh and aged smoke observations (from 0.13 ± 0.01 to 0.17 ± 0.02 ×10$^{-6}$ m) at several AERONET stations reported in Ansmann et al. (2021). However, Ansmann et al. (2012) also applied a high value of 0.24 ± 0.02 ×10$^{-6}$ m for the mass concentration retrieval of smoke aerosols (fine mode) when studying lofted layers containing desert dust and biomass burning smoke. It is hard to distinguish between smoke and urban haze aerosols, as they are often small (with size up to about 1 μm in radius) and quasi-spherical aerosols. Further, the characteristic conversion factors are in the similar value range. For examples, Ansmann et al., (2011) reported a conversion factor of 0.18 ± 0.02 ×10$^{-6}$ m for the central European haze; Mamali et al., (2018) found a factor of 0.14 ± 0.02 ×10$^{-6}$ m for continental/pollution particles over Cyprus; Mamouri et al., (2017) computed a factor of 0.30 ± 0.08 ×10$^{-6}$ m for continental aerosol pollution over Germany.

Air mass sources of aerosols on 6 June were investigated by the backward trajectory analysis (HYSPLIT model). It shows that some of the particles were coming from the forest fire in Canada region, while part of them were transported from Poland where urban haze could have been with smoke aerosols (e.g., Fig. 6). The aerosol subtype products (version 4.20) from CALIPSO when the orbit passing over Poland on 3 June (orbit from UTC 11:44 to 11:58) and 4 June (orbit from UTC 01:18 to 01:31) indicate the presence of polluted continental/smoke and polluted dust.

Consequently, it is possible that European pollution was mixed with Canadian smoke aerosols on 6 June in the fine-mode particles. Hence, the retrieved conversion factors cannot perfectly describe the smoke. However, in this section we still assume these factors reflect the smoke, so as to do the comparison analysis of estimated mass concentration from Polly$^{XT}$ and CL51.

[Figure]

**Figure 6. Five-day backward trajectories from the HYSPLIT model (a) in Frequency option, and (b) in Ensemble option, ending at 6h UTC on 6 June 2019 for Kuopio, Finland. The end location of the air mass is at 1.5 km agl in the range-transported plume.**

"

**P10, lines 295-300, this is a 'pure' speculation about dust (only fine-mode dust, no coarse mode dust), is my feeling. On the other hand, the enhanced depolarization ratio can easily be explained by non-spherical smoke. Already small deviations from the ideal spherical shape causes depolarization as Gialitaki et al., ACP, 2020 shows.**

**P10, lines 301-318, These paragraphs do not make any sense. I would remove this part. It is pure speculation. Sure, you may have fine dust, but without the presence of any coarse dust? Is that possible? And again, non-spherical smoke is a convincing argument for the enhanced depol values.**

**P11, L319-334. All this should be removed, just speculation, simply not convincing! Impossible, to accept that as a reviewer!**

Thank you for the suggestions. We have changed this section as "3.3 Comparison with MERRA-2 model - wildfire smoke and dust aerosol mixture".
The motivation for this analysis came from the MERRA-2 simulation results which indicated the presence of dust in the smoke plume. Therefore, we had to check how the inclusion of dust in our retrievals would affect the correspondence with the simulated mass profiles. The main conclusion from this exercise is that the mass retrievals are not that sensitive to aerosol types, which is good news for ceilometer retrievals as we are not able to consider mixed aerosol layers with ceilometer data only. Therefore, we feel that this discussion is a valuable addition to the manuscript and we would like to keep it. We added AIRS "Dust score" as additional dust information.

In section 3:
"
The backward trajectory analysis was performed using the HYSPLIT model. The analysis shows that particles in the SPoI had travelled about seven days from the forest fire sources (MODIS, 2019) in western Canada to North Europe (Fig. 4). The AIRS dust score map (https://airs.jpl.nasa.gov/map/, last access: 1 July 2021) also showed some dust presence in North America on 30 May.
"
We have clarified this in section 3.3:

"

The mass concentrations from MERRA-2 model data are used for the comparison with the lidar retrievals. An interesting feature in the MERRA-2 simulation results is the presence of dust in the SPoI. The contribution of dust to the total AOD is very low (much lower than the carbon optical depth), indicating that the dust particles are in the fine mode. However, the dust contribution to the total mass concentration is non-negligible. Low values of lidar-derived depolarization ratio suggest no significant presence of non-spherical particles, but in principle, a small amount of dust could be mixed with the smoke. It is possible that there are biomass burning aerosols and fine dust aerosols in the SPoI, as only fine dust particles should be able to remain long enough in the atmosphere to be transported from North America to Kuopio. Furthermore, the air masses in SPoI passed by the area in North America where dust was present (shown by the AIRS data).

…

In order to check how the inclusion of dust (as indicated by MERRA-2) would affect the mass concentration estimations, we assume that there were wildfire smoke and fine dust aerosol mixture in the SPoI. The POLIPHON (Polarization lidar photometer networking) method (Mamouri and Ansmann, 2014, 2017) was applied to separate fine dust (particles with radius < 500 nm) and biomass burning aerosols for the SPoI.

"

Our NASA colleagues V. Buchard and A.S. Darmenov provide the re-analysed MERRA-2 data for this manuscript, suggesting the fine dust presence. Thus, we mainly consider the fine dust and smoke mixture in the manuscript. In order to check if our conclusion is still valid when there is coarse mode dust mixture, we have also performed analysis considering coarse mode dust mixture. In the following figure, we estimate the mass concentrations by considering coarse mode dust (dc, dashed lines) mixture or fine mode dust (df, dotted lines). The total mass concentration is higher (of ~20–30 %) when considering coarse mode dust, but still within the uncertainty range. We didn't add this figure in the manuscript, but we have added discussions as follows in the revised version.

[Figure]

"

For the example given in Fig. 8, the fine dust contributes ~ 13 % to the extinction in the SPoI, whereas its mass concentration contributes ~ 32 % (method #1) or ~ 23 % (method #2) to the total mass concentration. However, the derived total mass concentration considering a fine dust and smoke mixture is only ~ 18 % (method #1) or ~ 4 % (method #2) higher than one assuming smoke particles only. The inclusion of a dust mixture results in slightly higher estimated mass concentration values, with a difference negligible considering the uncertainties. We have also performed POLIPHON considering coarse mode dust mixture; higher (~20–30 %) total mass concentrations were retrieved but still within the uncertainty range. The aged smoke aerosols may also introduce enhanced depolarization ratios. If we use a bigger value (e.g., 0.05) instead of 0.03 as the smoke depolarization ratio in POLIPHON, the dust impacts on the mass concentration estimations are even smaller. Hence, the mass estimations of the SPoI considering only smoke are good enough even if the plume contains small amount of dust.

Similar conclusion can also be applied to ceilometer observations. It is not possible to perform the aerosol separation using ceilometer data alone, as no depolarization information is available at this wavelength. For this instrument, only one aerosol type should always be assumed in the layer of interest, which then imparts an additional bias when estimating the mass concentration. However, we have shown in this section that ceilometer

observations for mass retrievals can be used even without exact knowledge on the composition of the smoke plume in the troposphere.

"

**To continue: Surface (in situ) observation cannot be used when discussing lofted layers. And the in situ measured aerosol values are most probably enhanced because of the advected central European haze. So, the final paragraph in Sect. 3 (before Sect. 3.1) makes no sense, and should be skipped.**

Thank you for the suggestion. We used in situ measurements to show that the increase in the aerosol mass concentration and absorption and scattering coefficients at the ground level was observed only after 6 June, after the deposition of the aerosol particles from this layer to the ground. The in situ measurements also demonstrated a high black carbon content and an increase in the brown carbon fraction at the ground level, indicating aerosol of biomass burning origin on 6th, when we used the AERONET factor.

But we agree with the referee that those in situ measurements don't offer much information in this paper, and we have removed all related parts (one paragraph in section 2.1, the final paragraph in Sect. 3, some sentence in the conclusion, and the figure of in situ measurements) in the revised version.

**P8, line 234: Sedimentation of large particles is not a good argument here. Smoke particles always show a pronounced accumulation mode, so difference in falling speed is low, when coarse mode particles are absent. Particle aging is more likely. Smoke aging process mainly occur in the first 36-48 hours after emission, and afterwards aging is slow. At the end of this aging process, the particles are usually spherical or almost spherical in shape. Particles show an almost perfect core-shell structure (coating, OC material) and the shell is often liquid at lower heights. And the probability that smoke particle are glassy (not perfectly round) increase with decreasing temperature. That could also be a reason that you saw a decreasing trend in the depolarization values with decreasing height.**

Thank you for the comment. We made modifications in the revised version:

"

The smoke particles caused slightly enhanced particle linear depolarization ratios (PDR) at 355 nm (532 nm) with a mean value of $0.08 \pm 0.02$ $(0.05 \pm 0.01)$ in the smoke layer, suggesting the presence of partly coated soot particles or particles that have mixed with a small amount of dust or other non-spherical aerosol type. The layer-mean PDR at 355 nm (532 nm) decreased during the day, from ~ 0.11 (0.06) in the morning to ~ 0.05 (0.04) in the evening. The decrease of the PDR with time could be linked to the particle aging and related changed in the smoke particle shape properties, as stated by Baars et al. (2019). The relative humidity (RH) profiles from GDAS1 data showed low values in the lower atmosphere (<60 % below 6 km) before 15h UTC, and even lower RH (<40 %) at the SPoI altitude. RH slightly increased in the evening. The signal in the 407 nm Raman-shifted channel was used to determine the water vapor mixing ratio profile during night-time, showing that the layer-mean RH changed from ~ 27 % at 19 h to ~ 38 % at 23 h, which was associated with the advection of a moister air mass with a water vapor mixing ratio close to 1–3 g kg$^{-1}$. The smoke particles were dry, and then captured water vapor in the atmosphere during the evening. The decreasing temperature and increasing RH also increase the probability that smoke particles become glassy. The depolarization ratios of aged biomass burning aerosols (originating from Canada and/or North America) reported in the literature (Table 1) range from 0.01 to 0.11 (0.01 to 0.08) at 532 nm (355 nm). More information of the aged smoke from other regions can be found in the literature review by Adam et al. (2020, see the Supplement).

"

**Figure 6: If you include a 40 or 50% uncertainty bar to the mass concentration values, you do not need to speculate about any dust contribution!**

We agree, so we have removed the Fig.6c as it shows only few information. We have added a new subfigure to show the difference on the estimated mass concentrations.

**In general, I miss uncertainty bars in Figures 6 and 7. Not many, but at least one or two per lidar and ceilometer profile!**

We have added the uncertainty bars.

**Now we need conclusions: One conclusion should deal with the question: What is now the value of the ceilometer? The ceilometer is able to detect smoke layers even in the middle to upper troposphere? With what overall uncertainty? What about mass retrieval from ceilometer observations? Possible? Yes or no? Conversion factors for 910 nm are not available. How to proceed? ..with 910/532 nm smoke backscatter color ratios? ..to convert 910 backscatter into 532 nm backscatter for which smoke conversion factors are available.**

Thank you for the very useful suggestions. We have added the colour ratio (or backscatter-related Ångström exponent) method as a new method #1 as a sub-section. Please check our previous replies. In the conclusions we have also emphasized our goal and replied to the referee's questions.
"

Two methods, based on a combined lidar and sun-photometer approach (based on AERONET products), were applied to both Polly$^{XT}$ and CL51 data for estimating mass concentrations: method #1, measured backscatter coefficients were converted to backscatter coefficients at 532 nm by corresponding measured backscatter-related Ångström exponent, and then be applied to estimate the mass concentrations; method #2, mass concentrations were estimated from measured backscatter coefficients at each wavelength (355, 532, 1064 nm from Polly$^{XT}$, and 910 nm from CL51) independently. A difference of ~ 12 % or ~ 36 % was found between Polly$^{XT}$ and CL51 estimated mass concentrations using method #1 or #2, showing that ceilometers are potential tools for mass concentration retrievals with ~ 50 % uncertainty, but with great spatial coverage. The retrieved mass concentration profiles were also compared with MERRA-2 aerosol profiles, where we considered and analysed two scenarios in the SPoI – 1) only smoke particles and 2) mixture of fine dust and smoke aerosols, and reported with the corresponding uncertainties. The inclusion of dust in the retrieved mass concentration is negligible considering the uncertainties; which indicates that ceilometer observations for mass retrievals can be used even without exact knowledge on the composition of the smoke dominant aerosol plume in the troposphere. We demonstrated the potential of the Vaisala CL51 ceilometer to contribute to atmospheric aerosol research in the vertical profile (e.g., to monitor smoke in the troposphere), from sensor-provided attenuated backscatter coefficient to particle mass concentration.
"

---

## Referee Report (RR1)

The manuscript has improved after the first revision. However, I have a few comments / suggestions which relates to aspects which were not addressed properly or to some unclear expressions or misspelling. I refer mainly to my previous comments as the other reviewers may comment on their points. In bold, the text from the manuscript.

I suggest the publication of this manuscript after addressing all the points raised by Reviewers after the second revision.

General statement:

I was wondering why a common smoothing range was not used for both lidar and ceilometer. Thus, 12 bins smoothing for lidar and 9 bins smoothing for ceilometer would have given the same effective resolution of 90m.

Line 25: changed should be change

Lines 27-29: sentence is not clear:

**A complete ceilometer data processing for a Vaisala CL51 is presented, including the water vapor correction for high latitude for the first time, from sensor provided attenuated backscatter coefficient to particle mass concentration.**

Maybe:

A complete ceilometer data processing for a Vaisala CL51 is presented (including the water vapor correction for high latitude for the first time) and the estimation of the particle mass concentration from sensor provided attenuated backscatter coefficient.

Please revise and state what is performed 'for the first time'.

Lines 98-111: references
Please add:
Tsaknakis et (2011) for ceilometer capacity of measuring smoke layers and dust layers (Atmos. Meas. Tech., 4, 1261–1273, 2011, www.atmos-meas-tech.net/4/1261/2011/)
Cazorla et al (2017) for near real time monitoring of a dust outbreak (Atmos. Chem. Phys., 17, 11861–11876, 2017, https://doi.org/10.5194/acp-17-11861-2017)
Adam et al (2016) for operational ceilometer network for pollution events monitoring (EPJ Web of Conferences, 119, 27007, 2016, ILRC 27, DOI: 10.1051/epjconf/201611927007,)
Dionisi et al (2018) for ceilometer estimates of mass concentration (Atmos. Meas. Tech., 11, 6013–6042, 2018, https://doi.org/10.5194/amt-11-6013-2018).

Line 128-129:
**This study reports, for the first time, a quantitative comparison study for Raman lidar and ceilometer observations of smoke particles.**

It is not clear what quantitative comparison for smoke particles means. Do you refer to smoke mass concentration? Please state it. Tsaknakis et al showed comparisons for attenuated backscatter from Raman lidar and Vaisala ceilometer. Mass concentration comparison between model and ceilometer (derived) is shown by Dionisi et al. No specific case of smoke only is discussed though.

Lines 131-132

E-profile is the good example of monitoring smoke, dust and other aerosol layers. I don't know how many papers are published. E.g. Vaughan et al, 2019.

Line 162:

When talking about radiosonde sounding, please cite Weigner et al 2019 (Atmos. Meas. Tech., 12, 471–490, 2019, https://doi.org/10.5194/amt-12-471-2019)

Line 165: GDAS1

It is not discussed the uncertainty of the water vapor transmission was assumed, using GDA1 for obtaining the water vapor number concentration. Figure 1 shows the uncertainty in the backscatter coefficients but we don't know what the input uncertainty in water vapor transmission was. The use of RH derived from Raman water vapor channel was no mentioned (as PollyXT provided it). From RH one derives AH and then number density (e.g. Bedoya-Velasquez et al 2021 (Atmospheric Research 250 (2021) 105379). However, Bedoya-Velasquez uses MWR to get T and RH.

Please add for reference Bedoya-Velasquez et al 2021 (Atmospheric Research 250 (2021) 105379) for water vapor correction.

Line 273:

In Haarig it is 82 $\pm$ 27. Please correct (also in Table 2).

Line 287:

If you assume that the pollen is well mixed in PBL, please mention it.

Lines 405-406

I wonder if the large difference at 14-16h UTC can be due to an inaccurate estimate of the water vapor transmission term in ceilometer retrieval. Usually, the water vapor amount is higher during day time.

---

## Author Response (AR2)

**Response to Referee #1**

**For referee comment no.2**

Thank you for carefully checking the manuscript and providing useful suggestions to improve the paper. The replies to the referee comments are given below. The referee comments are in blue with our responses in black. The sentences in the manuscript are between the quotation marks, with the modifications in red.

The manuscript has improved after the first revision. However, I have a few comments / suggestions which relates to aspects which were not addressed properly or to some unclear expressions or misspelling. I refer mainly to my previous comments as the other reviewers may comment on their points. In bold, the text from the manuscript.

I suggest the publication of this manuscript after addressing all the points raised by Reviewers after the second revision.

General statement:

I was wondering why a common smoothing range was not used for both lidar and ceilometer. Thus, 12 bins smoothing for lidar and 9 bins smoothing for ceilometer would have given the same effective resolution of 90m.

In our current data processing program, we used odd number of bins as the sliding window for the vertical smoothing. In our future data analysis, we will improve the program so that we can used both odd and even number of bins. In this manuscript we mainly consider the layer mean values, thus we think the current vertical smoothing don't have significant impacts on the results.

Line 25: changed should be change

The correction has been done.
"
  this decrease with time could be linked to the particle aging and related changes in the smoke particle shape properties.
"

Lines 27-29: sentence is not clear:

**A complete ceilometer data processing for a Vaisala CL51 is presented, including the water vapor correction for high latitude for the first time, from sensor provided attenuated backscatter coefficient to particle mass concentration.**

Maybe:

A complete ceilometer data processing for a Vaisala CL51 is presented (including the water vapor correction for high latitude for the first time) and the estimation of the particle mass concentration from sensor provided attenuated backscatter coefficient.

Please revise and state what is performed 'for the first time'.

Thank you for the suggestion. We made modifications for the clarity:
"
  A complete ceilometer data processing for a Vaisala CL51 is presented from sensor provided attenuated backscatter coefficient to particle mass concentration (including the water vapor correction for high latitude for the first time).
"

Lines 98-111: references

Please add:

Tsaknakis et (2011) for ceilometer capacity of measuring smoke layers and dust layers (Atmos. Meas. Tech., 4, 1261–1273, 2011, www.atmos-meas-tech.net/4/1261/2011/)

Cazorla et al (2017) for near real time monitoring of a dust outbreak (Atmos. Chem. Phys., 17, 11861–11876, 2017, https://doi.org/10.5194/acp-17-11861-2017)

Adam et al (2016) for operational ceilometer network for pollution events monitoring (EPJ Web of Conferences, 119, 27007, 2016, ILRC 27, DOI: 10.1051/epjconf/201611927007,)

Dionisi et al (2018) for ceilometer estimates of mass concentration (Atmos. Meas. Tech., 11, 6013– 6042, 2018, https://doi.org/10.5194/amt-11-6013-2018).

Thank you for the suggestion, we made modifications in the introduction as:
"

Ceilometer measurements have been used in several aerosol studies even though the instruments were originally designed to measure cloud heights. From an Arctic station, Mielonen et al. (2013) reported ceilometer observations of biomass burning plume heights from the 2010 Russian wildfires in northern Finland. Tsaknakis et al. (2011) present an inter-comparison of lidar and ceilometer measurements under different atmospheric conditions (urban air pollution, biomass burning and Saharan dust event), showing good agreements in determining the mixing layer height and the attenuated backscatter coefficient. Cazorla et al. (2017) present the implementation of procedures to manage the Iberian Ceilometer Network (ICENET) for monitoring aerosol characterization for near real time, which has been tested during a dust outbreak. Ceilometer measurements of the German Weather Service (DWD) network (http://www.dwd.de/ceilomap, last access: 20 July 2021) were employed to follow the progression of the volcanic ash layer (Emeis et al., 2011), and to visualise the dispersion and temporal development of the North American smoke plumes (Trickl et al., 2015). Vaughan et al. (2018) showed how a dense network of lidars and ceilometers in UK tracked the evolution of Canadian forest fire smoke. Adam et al. (2016) demonstrated that the operational ceilometer network of the Met Office can also provide valuable information for monitoring pollution events. Huff et al. (2021) demonstrated that ceilometers in the Unified Ceilometer Network (UCN, https://alg.umbc.edu/ucn/, last access: 20 July 2021) can verify and track smoke plume transport from a prescribed fire, in Maryland. Calibrated ceilometer profiles were also used as a tool to evaluate the aerosol forecasts by the European Centre for Medium-Range Weather Forecasts (ECMWF) Integrated Forecasting System aerosol module (IFS-AER) (Flentje et al., 2021). Dionisi et al. (2018) proposed a model-assisted methodology to retrieve key aerosol properties (such as extinction coefficient, surface area, and volume) from ceilometer measurements, under continental conditions; the good performances of that approach suggest that ceilometers can provide quantitative information for operational air quality and meteorological monitoring.  In order to analyse to what extent the existing ceilometer infrastructure could do in case of smoke monitoring, we performed a comparison study using an advanced Raman lidar, a ceilometer, and model data.

"

Line 128-129:

**This study reports, for the first time, a quantitative comparison study for Raman lidar and ceilometer observations of smoke particles.**

It is not clear what quantitative comparison for smoke particles means. Do you refer to smoke mass concentration? Please state it. Tsaknakis et al showed comparisons for attenuated backscatter from Raman lidar and Vaisala ceilometer. Mass concentration comparison between model and ceilometer (derived) is shown by Dionisi et al. No specific case of smoke only is discussed though.

Thank you for the comment, we made modifications for the clarity:
"

This study reports, for the first time, a quantitative comparison study of mass concentration estimates of smoke particles, for Raman lidar and ceilometer observations .

"

Lines 131-132

E-profile is the good example of monitoring smoke, dust and other aerosol layers. I don't know how many papers are published. E.g. Vaughan et al, 2019.

E-profile provide the near-real time quicklook of attenuated backscatter, which is good way for monitoring smoke plume location. But mass retrievals are likely even more valuable as they provide information on the mass load of the plume which is important information for example for aviation. We made modifications for the clarity:
"

Moreover, we demonstrate the usefulness of a Vaisala ceilometer to monitor smoke (in terms of quantitative information on the aerosol load) in the troposphere; the potential for mass concentration retrieval from ceilometer observations is also discussed.

"

Line 162:

When talking about radiosonde sounding, please cite Weigner et al 2019 (Atmos. Meas. Tech., 12, 471–490, 2019, https://doi.org/10.5194/amt-12-471-2019)

We have added the reference.
"

Aiming at the observations of water vapor profiles, the most used and well-established measurement method is radiosonde sounding (Wiegner et al., 2019).

"

Line 165: GDAS1

It is not discussed the uncertainty of the water vapor transmission was assumed, using GDAS1 for obtaining the water vapor number concentration. Figure 1 shows the uncertainty in the backscatter coefficients but we don't know what the input uncertainty in water vapor transmission was. The use of RH derived from Raman water vapor channel was no mentioned (as PollyXT provided it). From RH one derives AH and then number density (e.g. Bedoya-Velasquez et al 2021 (Atmospheric Research 250 (2021) 105379). However, Bedoya- Velasquez uses MWR to get T and RH.

Please add for reference Bedoya-Velasquez et al 2021 (Atmospheric Research 250 (2021) 105379) for water vapor correction.

Thank you for the comment. In this study, we used GDAS T and RH to derive absolute humidity and then the water vapor number density $n_w$, as the Raman lidar RH measurements were only available at night-time, and no nearby radiosonde measurements were available. Nevertheless, we have compared GDAS RH profiles with Raman lidar RH profiles, good agreements were found.
In the following figure we show for the case in Fig. 1 of the manuscript:
        left: the RHs from GDAS (blue, at 21h UTC) and measured by PollyXT (orange, 20-22h UTC),
        middle: derived water vapor number density $n_w$ from RH of GDAS or PollyXT, separately.
        right: calculated squared effective water vapor transmissions using RH of GDAS or PollyXT.
Note that we used GDAS temperature profile for the calculations.

[Figure]

We add in the manuscript:

"

The uncertainties range due to wrong assumptions of $\lambda_0 \pm 2$ nm is given by the horizontal lines in Fig. 1. The uncertainties in backscatter coefficients of the analytical solution were also shown by dashed lines. Bedoya-Velásquez et al. (2021) applied a water vapor correction method on the CL51 ceilometer measurements, based on the one proposed by Wiegner and Gasteiger (2015). They have also studied the sensitivity of the aerosol retrievals to the use of modelled temperature and absolute humidity from HYSPLIT to correct water vapor absorption, instead of the co-located Microwave radiometer measurements: it leads to errors in the pre-processed range-corrected signals up to 9 %, and in particle backscatter coefficients up to 2.2 %. Thus, an extra uncertainty should be considered as GDAS data were used for the water vapor correction. We cannot quantify the error in GDAS temperature. Nevertheless, Polly$^{XT}$ measured relative humidity (RH) were applied for the comparison, and good agreements were found. The relative difference on the squared effective water vapor transmissions using RH profiles of GDAS or Polly$^{XT}$ is less than 2 % for the case in Fig. 1. The input uncertainty in water vapor transmissions due to the use of modelled input was not taken into account in this study, as there was no means to quantify the value. As the water vapor contribution cannot be neglected at Kuopio during summer, the water vapor corrections have been applied to CL51 data in this study.

"

Line 273:

In Haarig it is 82 ± 27. Please correct (also in Table 2).

Thank you for pointing it out. We used lidar ratio values from table 3 of Haarig et al. which were obtained from the BERTHA measurements. But now we changed to values from table 1 of Haarig et al. which were mean values with all three lidars. We corrected the values in text and in tables:

"

A value of 82 sr for LR, as measured at 1064 nm (82 ± 27 sr in Haarig et al., 2018), was assumed as being appropriate for use at 910 nm in this study.

"

in Table 1:

| | Lidar ratio (sr) | | | PDR (%) | | Ångström exponent | | | R$_{eff}$ (µm) |
|---|---|---|---|---|---|---|---|---|---|
| | 355 | 532 | 1064 | 355 | 532 | EAE 355/532 | BAE 355/532 | BAE 532/1064 | |
| Haarig et al. (2018) | 45 ± 5 | 68 ± 9 | 82 ± 27 | 2 ± 4 | 3 ± 2 | 0.9 ± 0.5 | 2.1 ± 0.6 | 0.8 ± 0.3 | 0.17 ± 0.06 |

in Table 2:

| | Parameter | Wavelength | Value | References |
|---|---|---|---|---|
| Method #2 | Lidar ratio (sr) | 355 | 47 ± 5 | This study |
| | | 1064 | 82 ± 27 | Haarig et al. (2018) |
| | | 910 | 82 ± 27* | Haarig et al. (2018) |

Line 287:

If you assume that the pollen is well mixed in PBL, please mention it.

Thank you for the comments. We made modifications for the clarity:
"

Our in situ pollen measurements (more information about pollen instruments can be found in Bohlmann et al., 2021) shows high pine pollen loading at the ground (highest 2 h pollen concentrations were ~ 3000 m$^{-3}$ on 5 June and ~ 7000 m$^{-3}$ on 6 June). The pollen particles were well mixed in the boundary layer (below 2 km), causing strong backscattering together with high depolarization ratio at 532 nm with a clear diurnal cycle.

"

Lines 405-406

I wonder if the large difference at 14-16h UTC can be due to an inaccurate estimate of the water vapor transmission term in ceilometer retrieval. Usually, the water vapor amount is higher during day time.

Thank you for the idea! We looked into this as you are right that typically the water vapor amount is higher during daytime. But water vapor transmission can be ignored at PollyXT wavelengths and retrievals based on PollyXT observations also showed large differences when compared with the model profiles. Therefore, we think that the reason for the discrepancies originates from somewhere else than the water vapor transmission.